# Boosting Graph Anomaly Detection with Adaptive Message Passing

**Jingyan Chen, Guanghui Zhu**\*, **Chunfeng Yuan, Yihua Huang**
State Key Laboratory for Novel Software Technology, Nanjing University
chenjy@smail.nju.edu.cn, {zgh, cfyuan, yhuang}@nju.edu.cn

## Abstract

Unsupervised graph anomaly detection has been widely used in real-world applications. Existing methods primarily focus on local inconsistency mining (LIM), based on the intuition that establishing high similarities between abnormal nodes and their neighbors is difficult. However, the message passing employed by graph neural networks (GNNs) results in local anomaly signal loss, as GNNs tend to make connected nodes similar, which conflicts with the LIM intuition. In this paper, we propose GADAM, a novel framework that not only resolves the conflict between LIM and message passing but also leverages message passing to augment anomaly detection through a transformative approach to anomaly mining beyond LIM. Specifically, we first propose an efficient MLP-based LIM approach to obtain local anomaly scores in a conflict-free way. Next, we introduce a novel approach to capture anomaly signals from a global perspective. This involves a hybrid attention based adaptive message passing, enabling nodes to selectively absorb abnormal or normal signals from their surroundings. Extensive experiments conducted on nine benchmark datasets, including two large-scale OGB datasets, demonstrate that GADAM surpasses existing state-of-the-art methods in terms of both effectiveness and efficiency.

## 1 Introduction

Anomaly detection on a graph refers to identifying abnormal nodes, which has emerged as a crucial research field (Ma X, 2021; Ren et al., 2023; Wang L, 2021; Liu C, 2021), widely applied in various domains such as social networks (Cheng L, 2021; Yuan H, 2021), financial fraud detection (Huang X, 2022; Dou Y, 2020), and telecommunication fraud detection(Yang et al., 2019). The scarcity of labels and diversity of anomalies makes unsupervised graph anomaly detection (UGAD) an important, but non-trivial problem. There are two main types of abnormal nodes in graphs: contextual and structural anomalies (Liu et al., 2022a), as illustrated in Fig. 1. A surge of recent works (Ding et al., 2019; Fan et al., 2020; Bandyopadhyay et al., 2020; Liu et al., 2021; Yuan et al., 2021; Xu et al., 2022; Zhang et al., 2022) have explored various effective methods for UGAD. Due to the lack of labels, most of them are based on local inconsistency mining (LIM), with the intuition that establishing significant similarity between abnormal nodes and their neighbors is more difficult than normal nodes. Graph neural networks (GNNs) (Pan S, 2021; Thomas N. Kipf, 2017; P. Veliˇckoviˊc & Bengio, 2018) are utilized for node representation learning, reconstruction error or similarity discriminator is used to measure the inconsistency between the node and its neighbors, and higher inconsistency refers to a larger anomaly score.

Despite the success of existing methods, an important but overlooked problem is **the conflict between the message passing of GNNs and the LIM intuition.** Specifically, the feature aggregation operation of GNNs tends to make connected nodes similar (Chai et al., 2022; Li et al., 2019a; Wu et al., 2019; Balcilar et al., 2021), which implicitly enhances the consistency between the node and its neighbors and makes anomalies indistinguishable. A naive solution is to employ models without message passing, such as using MLP (Ramchoun et al., 2016), to enable conflict-free LIM. However, the graph structure contains valuable anomaly signals. For instance, densely connected subgraphs

---

\*Corresponding Author

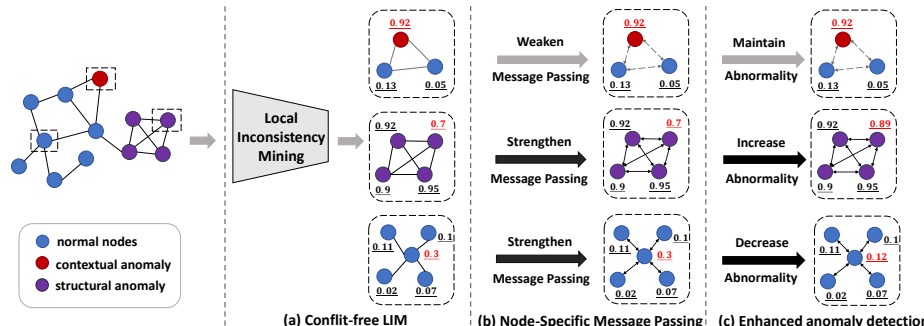

Figure 1: Two types of anomalies and the guide map of designing UGAD methods. Contextual anomalies are nodes with significantly different attributes from their neighbors. Structural anomalies are densely connected nodes with abnormal links in contrast to sparsely connected regular nodes.

may suggest nodes belonging to structural anomalies. Simultaneously, the smoothing effect of message passing reduces the anomaly degree in normal nodes by aligning them with their neighbors. Therefore, completely discarding the message passing would be unwise.

Meanwhile, various approaches (Li et al., 2019b; Fan et al., 2020; Pei et al., 2021) have been developed to minimize the damage of message passing on LIM-based anomaly detection. These methods aim to enhance the local salience of anomalies and produce distinguishable representations, often utilizing attention mechanisms to reduce the message passing between anomalous nodes and their neighbors. But on the one hand, such approaches only apply to discover contextual anomalies that have inherently inconsistent features compared to neighboring nodes. In contrast, structural anomalies tend to form clusters that share similar attributes and abnormal behaviors. Striving to block the message passing and deliberately maintaining clusters of anomalies inconsistent with their surroundings conflict with the inherent nature of structural anomalies. On the other hand, the fundamental conflict still remains. The core reason why message passing is harmful to anomaly detection is the adoption of local inconsistency as a measure of anomaly signal. Existing semi-supervised approaches (Chen et al., 2022; Liu Y, 2021; Gao et al., 2023; Dou Y, 2020) have achieved performance boost through establishing fine-grained message passing and rational anomaly mining approaches. Nevertheless, there remains a significant gap in how to design an anomaly mining approach that seamlessly integrates with message passing in unsupervised scenarios.

The above analysis inspires us that designing an advanced UGAD method is a journey of handling message passing, which is guided by a map that consists of several key components as illustrated in Fig. 1: (1) Ensuring conflict-free LIM through elaborate control of message passing. (2) Achieving fine-grained control of node-specific message passing with its neighbors in the absence of labels. (3) Designing new learning objective and anomaly mining approach beyond LIM, harnessing the utility of message passing for enhanced anomaly detection.

To this end, we propose **GADAM**[1] (**G**raph **A**nomaly **D**etection with **A**daptive **M**essage passing), a clear and flexible two-stage framework that performs LIM and message passing decoupled. In the first stage, we propose an MLP-based contrastive method without message passing for LIM, which enables LIM to more effectively and efficiently identify anomalous nodes and produce anomaly scores from a local perspective. In the second stage, we convert the learning objective to a binary classification task, utilizing local anomaly scores as pseudo-labels. Concretely, two high-confidence normal and abnormal node sets are established using the local anomaly scores from the first stage. Subsequently, we define the learning objective to discriminate the alignment between nodes and the global normal context, i.e. average of high-confidence normal nodes, and generate anomaly scores from a global view. The second stage will be semi-supervisedly trained, guided by pseudo-labels, and facilitated by an adaptive message passing scheme. To achieve adaptive message passing for nodes with distinct roles, we further propose an innovative hybrid attention mechanism that takes into account both the anomaly score difference and the feature similarity with its neighbors. The shift in the perspective of anomaly mining, combined with the design of the hybrid attention mechanism, enables message passing to enrich anomaly mining, akin to how feature aggregation in GNNs enhances classification tasks (Huang et al., 2020; Wang & Zhang, 2022; Zhu, 2005). The anomaly scores from the two stages will be combined to provide a holistic measure of node anomaly.

---

[1]GADAM is available at https://github.com/PasaLab/GADAM

The main contributions of our paper are summarized as follows:

- We analyze the dilemma faced by current UGAD methods when handling message passing. A novel MLP-based contrastive method is introduced to disentangle message passing from LIM, enabling more effective and efficient LIM.

- We introduce a novel anomaly mining approach from a global view, which utilizes local anomaly scores as pseudo-labels and models anomaly detection as a binary classification task to facilitate message passing, and enhance the effectiveness of anomaly detection.

- A well-designed hybrid attention mechanism is proposed, which takes into account both a node's anomaly score difference and the feature similarity with its neighbors, enabling more precise and adaptive message passing.

- Extensive experimental results on nine benchmark datasets, including seven datasets with injected synthetic anomalies and two datasets with organic anomalies, demonstrate that our method achieves state-of-the-art performance compared with a range of baselines. Moreover, GADAM shows superiority in both runtime and GPU overhead.

## 2  BACKGROUND

### 2.1  UNSUPERVISED GRAPH ANOMALY DETECTION

Let $\mathcal{G} = (\mathcal{V}, \mathbf{A}, \mathbf{X})$ be the input graph where $\mathcal{V} = \{v_1, v_2, ..., v_N\}$ is the set of $N$ nodes, $\mathbf{A} \in \mathbb{R}^{N \times N}$ is the adjacent matrix and $\mathbf{X} \in \mathbb{R}^{N \times F}$ is the node attribute matrix where the $i$-th row $\mathbf{X}[i, :]$ denotes the attribute of node $v_i$. UGAD aims to learn an anomaly score vector $\mathcal{S} \in \mathbb{R}^N$ to indicate the anomaly degree for every node, where a larger element $s_i$ means a higher abnormality for node $v_i$.

### 2.2  CONTRASTIVE LEARNING FOR UGAD

CoLA (Liu et al., 2021) is the first to leverage contrastive learning for LIM, and its process can be divided into four parts. **(1) Contrastive instance pairs construction:** For a given node $v_i$, CoLA samples an subgraph $\mathcal{G}_p^i = (\mathcal{V}^{(i)}, \mathbf{A}^{(i)}, \mathbf{X}^{(i)})$ with size $|\mathcal{V}^{(i)}| = M$ by random walking with the center node $v_i$, and construct the positive instance pair $\langle v_i, \mathcal{G}_p^i \rangle$. Another subgraph $\mathcal{G}_n^i = (\mathcal{V}^{(j)}, \mathbf{A}^{(j)}, \mathbf{X}^{(j)})$ is sampled with the center node $v_j (i \neq j)$ to form the negative instance pair $\langle v_i, \mathcal{G}_n^i \rangle$. $\mathcal{G}_p^i$ and $\mathcal{G}_n^i$ have the same size. **(2) Instance pair representation learning:** The central nodes $v_i, v_j$ are anonymized by setting their attributes to $\overrightarrow{0}$ in $\mathcal{G}_p^i$ and $\mathcal{G}_n^i$ respectively. Then, an $L$-layer graph convolutional network (GCN) with parameters $\mathbf{W}$ is used to obtain node embeddings, and a readout function is employed to yield the subgraph representation: $\boldsymbol{e}_p^i = \sum_{m=1}^{M} \frac{(\mathbf{E}_p^i)_m}{M}$, $\mathbf{E}_p^i = GCN(\mathcal{G}_p^i; \mathbf{W}, \phi)$. $\mathbf{E}_p^i$ represents node embeddings in $\mathcal{G}_p^i$ and $\phi$ is a activation function. The negative subgraph representation $\boldsymbol{e}_n^i$ is obtained in the same way on $\mathcal{G}_n^i$. Next, an $L$-layer feed forward network (FFN) with parameters $\mathbf{W}$ and activation function $\phi$ same as GCN is utilized to map node $v_i$ to the same feature space: $\boldsymbol{h}_i = FFN(\mathbf{X}[i, :]; \mathbf{W}, \phi)$. **(3) Similarity discriminating:** A discriminator with parameter $\mathbf{W}^{(\mathbf{d})}$ is used to evaluate the similarity between contrastive pairs: $Dis(\boldsymbol{h}_i, \boldsymbol{e}_{(p,n)}^i) = \sigma(\boldsymbol{h}_i \mathbf{W}^{(\mathbf{d})}(\boldsymbol{e}_{(p,n)}^i)^T)$, where $\boldsymbol{e}_{(p,n)}^i$ denotes $\boldsymbol{e}_p^i$ or $\boldsymbol{e}_n^i$. The similarity of positive instance pair $\langle \boldsymbol{h}_i, \boldsymbol{e}_p^i \rangle$ will be increased and the opposite for negative instance pair $\langle \boldsymbol{h}_i, \boldsymbol{e}_n^i \rangle$. CoLA is learned by minimizing the binary cross-entropy (BCE) loss:

$$\mathcal{L} = -\sum_{i=1}^{N} \log(Dis(\boldsymbol{h}_i, \boldsymbol{e}_p^i)) + \log(1 - Dis(\boldsymbol{h}_i, \boldsymbol{e}_n^i)) \tag{1}$$

**(4) Multi-round anomaly score computation:** At the inference stage, $R$ positive and negative subgraphs will be sampled for node $v_i$, forming an instances pool $(\mathcal{G}_p^{i,1}, ..., \mathcal{G}_p^{i,R}, \mathcal{G}_n^{i,1}, ..., \mathcal{G}_n^{i,R})$. The anomaly score is obtained through $R$-round evaluation to eliminate the randomness caused by subgraph sampling:

$$s_i = \frac{\sum_{r=1}^{R} Dis(\boldsymbol{h}_i, \boldsymbol{e}_n^{i,r}) - Dis(\boldsymbol{h}_i, \boldsymbol{e}_p^{i,r})}{R} \tag{2}$$

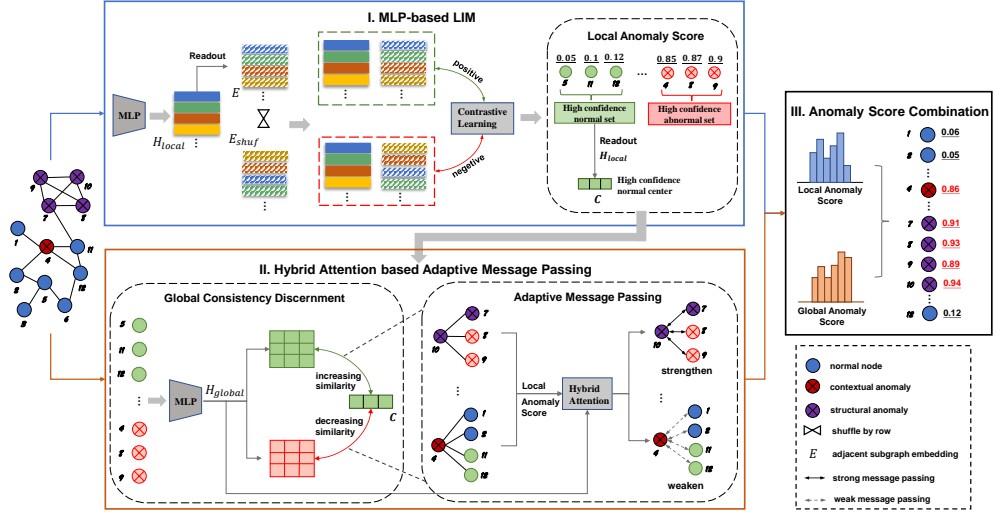

Figure 2: Overall workflow of GADAM.

## 3 THE PROPOSED METHOD

As shown in Fig. 2, the overall workflow of GADAM contains two stages: (1) **MLP-based LIM** employs MLP as encoder and extract local anomaly signals through contrastive learning, (2) **Hybrid Attention based Adaptive Message Passing** is designed to calculate the global anomaly score by discriminating the alignment between nodes and the global normal context, accompanied by adaptive message passing with neighbors. Next, we introduce each stage in detail.

### 3.1 MLP-BASED LOCAL INCONSISTENCY MINING

Considering that contrastive learning can directly target the anomaly detection objective and is friendly to minibatch training (Liu et al., 2021), we employ contrastive learning, along with the following proposed enhancements for more precise and efficient LIM:

(1) **MLP as encoder:** We utilize an $L$-layer MLP for all nodes with L2 normalization to obtain node embedding matrix $\mathbf{H}_{local}$, as in Eq.(3). Leveraging MLP instead of GNN as an encoder maintains the independence of node representations, thus avoiding conflicts between LIM and message passing.

$$\mathbf{H}_{local}^{(l)} = \sigma(\mathbf{H}_{local}^{(l-1)}\mathbf{W}^{(l-1)} + \mathbf{b}^{(l-1)}), \ \ \mathbf{H}_{local}^{(l)}[i,:] = \frac{\mathbf{H}_{local}^{(l)}[i,:]}{||\mathbf{H}_{local}^{(l)}[i,:]||_2}, \ \ \mathbf{H}_{local}^{(0)} = \mathbf{X} \quad (3)$$

(2) **Sampling-free contrastive pair construction:** We set the positive instance $\mathcal{G}_p^i$ as the complete adjacency subgraph of node $v_i$ instead of sampling. The embedding of $\mathcal{G}_p^i$ is computed by averaging the node embeddings in $\mathcal{G}_p^i$, and the subgraph embedding matrix $\mathbf{E}$ is obtained where $\mathbf{E}[i,:] = \boldsymbol{e}_p^i$ denotes the embedding of $\mathcal{G}_p^i$. Then, we shuffle $\mathbf{E}$ by row to get $\mathbf{E}_{shuf}$. For node $v_i$, the positive and negative instance pairs can be expressed as:

$$\text{pos} = \langle \boldsymbol{h}_i^{local}, \mathbf{E}[i,:]\rangle, \ \ \text{neg} = \langle \boldsymbol{h}_i^{local}, \mathbf{E}_{shuf}[i,:]\rangle \quad (4)$$

where $\boldsymbol{h}_i^{local} = \mathbf{H}_{local}^{(l)}[i,:]$. The proposed contrastive pair construction method can avoid the time overhead caused by graph sampling.

(3) **Parameter-free similarity discriminator:** We utilize the inner product as a similarity discriminator, which is equivalent to cosine similarity after L2 normalization:

$$Dis(\boldsymbol{h}_i^{local}, \boldsymbol{e}_p^i) = \boldsymbol{h}_i^{local}(\boldsymbol{e}_p^i)^T, \ \ Dis(\boldsymbol{h}_i^{local}, \boldsymbol{e}_n^i) = \boldsymbol{h}_i^{local}(\boldsymbol{e}_n^i)^T \quad (5)$$

(4) **Direct anomaly score calculation:** Based on the LIM intuition, we directly use the feature similarity of the positive instance pairs, i.e., the node and adjacent subgraphs, as the anomaly score:

$$s_i^{local} = -Dis(\boldsymbol{h}_i^{local}, \boldsymbol{e}_p^i) = -\boldsymbol{h}_i^{local}(\boldsymbol{e}_p^i)^T \quad (6)$$

CoLA computes anomaly scores through multiple rounds of evaluation, because of the randomization involved in subgraph sampling and the construction of instance pairs. In contrast, our approach obviates the necessity for sampling, with anomaly scores relying solely on positive instance pairs. This modification leads to more efficient computation.

We optimize the first stage using the loss function defined in Eq. (1), and produce an anomaly score vector $\boldsymbol{S}^{local}$ with Eq. (6). The aforementioned enhancements not only mitigate conflicts between LIM and message passing, but also result in a reduction in model parameters and the refinement of modules, consequently improving both the model's effectiveness and efficiency.

After the first stage, we sort $\boldsymbol{S}^{local}$ and take the nodes with the smallest $k_{nor}\%$ of local anomaly scores to form a high-confidence normal set $\mathcal{V}_n$, and a representative global normal context can be obtained by averaging nodes in $\mathcal{V}_n$: $\boldsymbol{C} = \frac{1}{|\mathcal{V}_n|}\sum_{i\in\mathcal{V}_n}\boldsymbol{h}_i^{local}$. We also select $k_{ano}\%$ of nodes with the largest local anomaly scores to form a high-confidence abnormal set $\mathcal{V}_a$. Two high-confidence sets and global normal context $\boldsymbol{C}$ are served as supervised signals in the second stage.

## 3.2 Hybrid Attention based Adaptive Message passing

In the second stage, we further propose two components aimed at achieving node-specific message passing and an effective anomaly mining approach beyond LIM, respectively: (1) **Adaptive message passing with hybrid attention** is used to determine the strength of message passing based on the local anomaly score differences and feature similarities between nodes and their neighbors. (2) **Global consistency discernment** is employed to capture anomaly signals based on the consistency of nodes with the global normal context.

### 3.2.1 Adaptive Message passing with Hybrid Attention

(1) **Adaptive message passing:** We first employ another MLP and follow Eq. (3) for all nodes to get $\mathbf{H}_{global}$ in the second stage. For each node $v_i$ at epoch $t$, the node embedding is a mixture of $\boldsymbol{h}_i^{global}$ (i.e., $\mathbf{H}_{global}[i,:]$) and the adjacency subgraph embedding $\boldsymbol{e}_p^i$. The coefficient of $\boldsymbol{e}_p^i$ (i.e., $\alpha_i^{(t)}$) that facilitates the message strength is determined adaptively with a hybrid attention mechanism:

$$
\begin{aligned}
\boldsymbol{h}_i^{mix} &= \alpha_i^{(t)} \cdot \boldsymbol{e}_p^i + (1 - \alpha_i^{(t)}) \cdot \boldsymbol{h}_i^{global} \\
\alpha_i^{(t)} &= attention(\boldsymbol{h}_i^{global}, \mathcal{G}_p^i)
\end{aligned}
\tag{7}
$$

(2) **Hybrid attention mechanism:** The attention mechanism is to assign higher message passing weights to nodes with similar roles. We leverage two types of attention, namely pre-attention and post-attention, which focuses on the local anomaly score difference and the feature similarity respectively. Two types of attention are weighted summed to form a hybrid attention mechanism to precisely determine the strength of message passing.

- Pre-attention: Pre-attention is based on the local anomaly score differences between a node and its surroundings. For a given node $v_i$, the differences are calculated by Eq. (8):

$$
\delta_i = |s_i^{local} - \frac{1}{|\mathcal{N}(i)|}\sum_{j\in\mathcal{N}(i)} s_j^{local}|
\tag{8}
$$

where $\mathcal{N}(i)$ denotes the node set of $\mathcal{G}_p^i$. We take the mean of the nodes in $\mathcal{V}_n$ as a benchmark for $\delta$, and $d_i$ be the deviation of $\delta_i$ for node $v_i$:

$$
d_i = \frac{|\delta_i - \mathbb{E}_{j\in\mathcal{V}_n}(\delta_j)|}{\mathrm{Var}_{j\in\mathcal{V}_n}(\delta_j)}
\tag{9}
$$

An activation function is applied to convert $d_i$ to a value between $[0,1]$ to obtain pre-attention:

$$
pre_i = 1 - \sigma(d_i)
\tag{10}
$$

Intuitively, normal nodes and structural anomalies tend to exhibit smaller $\delta$, owing to their higher consistency in anomaly scores with their surroundings. Consequently, these nodes will also have smaller $d$ and larger pre-attention scores, while the opposite for contextual anomalies.

- Post-attention: We further propose to leverage feature similarity between a node and its neighbors to obtain post-attention:

$$
post_i = \boldsymbol{h}_i^{global}(\boldsymbol{e}_p^i)^T
\tag{11}
$$

As contextual anomalies inherently possess features distinct from their neighbors, the post-attention effectively assigns them smaller message passing weights, while the opposite for normal nodes and structural anomalies. Thus, pre-attention and post-attention can work harmoniously.

- Dynamic weighted sum: Two types of attentions are dynamically weighted summed to get the final attention coefficient for each node $v_i$ at epoch $t$:

$$\alpha_i^{(t)} = \beta^{(t)} \cdot pre_i + (1 - \beta^{(t)}) \cdot post_i$$
$$\beta^{(t)} = \beta \times (1 - \frac{t}{T_{global}}) \tag{12}$$

where $T_{global}$ is the total training epochs of the second stage, and $\beta < 1$ determines the initial weight of pre-attention. Intuitively, as the training advances, the influence of pre-attention gradually diminishes, while the opposite for post-attention. This shift serves a dual purpose: (1) It prevents inherent errors in local anomaly scores from continuously misdirecting attention for nodes. (2) As training progresses, nodes naturally adapt to better align with their neighboring nodes, thereby requiring a gradual increase in the weight of post-attention.

The diversity of attention metrics, the collaborative synergy between the two types of attention, and the dynamic weighted strategy collectively empower the hybrid attention mechanism to facilitate precise node-specific message passing. Additionally, a visualization of the hybrid attention is available in Appendix E for better clarity and understanding.

### 3.2.2 GLOBAL CONSISTENCY DISCERNMENT

Inspired by Chen et al. (2022), anomalies tend to be more distant from global normal context $\boldsymbol{C}$ than normal nodes. Therefore, we enhance the similarity of the high-confidence normal set $\mathcal{V}_n$ with $\boldsymbol{C}$ while decreasing it for the high-confidence abnormal set $\mathcal{V}_a$ with the following loss function:

$$g_i = Dis(\boldsymbol{h}_i^{mix}, \boldsymbol{C}) = \boldsymbol{h}_i^{mix} \boldsymbol{C}^T$$
$$\mathcal{L} = -\frac{1}{|\mathcal{V}_n| + |\mathcal{V}_a|} \sum_{j \in \mathcal{V}_n} log(g_j) + \sum_{k \in \mathcal{V}_a} log(1 - g_k) \tag{13}$$

The global anomaly score is obtained from the similarity of the nodes and center $\boldsymbol{C}$: $s_i^{global} = -g_i$, and vector $\boldsymbol{S}^{global}$ is obtained. Ultimately, anomaly detection can be performed by combining two anomaly scores: $\boldsymbol{S} = (\boldsymbol{S}^{local} + \boldsymbol{S}^{global})/2$.

### 3.3 EFFICIENCY ANALYSIS

Assume an input graph has $N$ nodes and $\mathcal{E}$ edges, and $\omega$ is the average node degree. Let $f$ be the dimension of the MLP in GADAM, we analyze the time complexity of GADAM by considering the two main stages respectively. (1) For MLP-based LIM, the main time complexity is node representation learning which takes $\mathcal{O}(Nf)$. (2) For the second stage, the time complexity is mainly generated by node representation learning through MLP, which is $\mathcal{O}(Nf)$; and hybrid attention, which is $\mathcal{O}(N\omega)$ for pre-attention and $\mathcal{O}(N\omega + Nf^2)$ for post-attention. For anomaly score combination, the time complexity can be ignored. In summary, the overall time complexity is $\mathcal{O}((2f + f^2)N + 2\mathcal{E})$, which is linearly dependent on the number of nodes and edges in the graph. Besides the high computation efficiency, the parameters of GADAM are only the MLPs in two stages, which is memory friendly and helpful for handling large-scale data. We also analyze the baselines listed in §4.1 and compare them with GADAM, see Appendix D for more details.

## 4 EXPERIMENTS

### 4.1 EXPERIMENT SETUP

**Datasets.** To comprehensively evaluate our model, we use the following nine benchmark datasets: (1) Seven real-world datasets including five widely used benchmark datasets (Sen et al., 2008; Tang et al., 2008; Tang & Liu, 2009): Cora, Citeseer, Pubmed, ACM, and BlogCatalog. And two large scale OGB (Hu et al., 2020) datasets: ogbn-arxiv and ogbn-products. Since there are no organic anomalies in these graphs, we follow previous research (Liu et al., 2022a) to inject synthetic anomalies, which is common and widely used in UGAD. (2) Two real-world datasets that contain organic

Table 1: ROC-AUC comparison on nine benchmark datasets.(**First** Second) IMP: the average improvement of GADAM over the rest. OOM: out of memory even if the batch size is set to 32.

| Method | Cora | Citeseer | Pubmed | ACM | BlogCatalog | ogbn-Arxiv | ogbn-Products | Books | Reddit |
|---|---|---|---|---|---|---|---|---|---|
| DOMINANT | 0.8493 | 0.8391 | 0.8013 | 0.7452 | 0.7531 | OOM | OOM | 0.5012 | 0.5621 |
| AnomalyDAE | 0.8431 | 0.8264 | 0.8973 | 0.7516 | 0.7658 | 0.6214 | OOM | 0.5567 | 0.5454 |
| AdONE | 0.8561 | 0.8724 | 0.7952 | 0.7219 | 0.7314 | OOM | OOM | 0.5366 | 0.5015 |
| CoLA | 0.8801 | 0.8891 | 0.9535 | 0.7783 | 0.7807 | 0.8041 | OOM | 0.3982 | 0.5791 |
| ANEMONE | 0.9054 | 0.9239 | 0.9464 | 0.8802 | 0.8005 | OOM | OOM | 0.4341 | 0.5563 |
| SL-GAD | 0.8983 | 0.9106 | 0.9476 | 0.8538 | 0.8037 | OOM | OOM | 0.5655 | 0.5625 |
| CONAD | 0.7423 | 0.7145 | 0.6993 | 0.6849 | 0.6557 | OOM | OOM | 0.5224 | 0.5610 |
| Sub-CR | 0.9132 | 0.9310 | **0.9629** | 0.7245 | 0.8071 | OOM | OOM | 0.5713 | 0.5563 |
| ResGCN | 0.8479 | 0.7647 | 0.8079 | 0.7681 | 0.7852 | OOM | OOM | 0.5665 | 0.5012 |
| ComGA | 0.8840 | 0.9167 | 0.9212 | 0.8496 | 0.8030 | OOM | OOM | 0.5354 | 0.5682 |
| GADAM | **0.9556** | **0.9415** | 0.9581 | **0.9603** | **0.8117** | **0.8122** | **0.8499** | **0.5983** | **0.5809** |
| IMP | 11.21% | 10.50% | 10.98% | 24.54% | 5.99% | 15.85% | - | 16.88% | 5.98% |

anomalies: Books (Sánchez et al., 2013) and Reddit (Kumar et al., 2019; Wang et al., 2021). Detailed dataset description and anomaly injection approach can be found in Appendix A.

**Baselines.** We compare with three classes of methods regarding UGAD. The first family of baselines are autoencoder based methods: DOMINANT (Ding et al., 2019), AnomalyDAE (Fan et al., 2020), and AdONE (Bandyopadhyay et al., 2020). The second family are CL-based methods: CoLA (Liu et al., 2021), ANEMONE (Jin et al., 2021), SL-GAD (Zheng et al., 2021), CONAD (Xu et al., 2022), and Sub-CR (Zhang et al., 2022). The third are models that are particularly designed for handling indistinguishable anomalies caused by message passing: ResGCN (Pei et al., 2021) and ComGA (Luo et al., 2022). We provide more details for these models in Appendix B.3.

**Evaluation metrics.** We follow the extensive literature (Ding et al., 2021; Tong & Lin, 2011; Liu et al., 2022a) in UGAD to comprehensively evaluate model performance with three metrics: (1) ROC-AUC evaluates the comprehensive performance of both normal and abnormal samples. (2) Average Precision focuses more on abnormal samples. (3) Recall@k evaluate the top-k samples with high predicted anomaly scores. More details can be found in Appendix B.5.

**Implementation details.** For GADAM, we set $\beta = 0.9$ in Eq. (12) to determine the initial weight of pre-attention. We set $k_{ano} = 1, k_{nor} = 50$ for Pubmed, and $k_{ano} = 5, k_{nor} = 30$ for rest datasets to establish high confidence sets. More hyperparameter settings can be found in Appendix B.6. For the implementation of baselines, we use the PyGod (Liu et al., 2022b) library if they are available, otherwise use the source code provided in the original paper.

## 4.2 EFFECTIVENESS COMPARISON

We compare GADAM with representative baselines, and the mean ROC-AUC of five runs are reported in Tab. 1. The variance is omitted due to its negligible magnitude. Other results for AP and Recall@k metrics are deferred to Appendix C. By comparison, we have the following observations:

- GADAM demonstrates remarkable effectiveness, showcasing substantial improvements ranging from 5.98% to 24.54% in ROC-AUC across all datasets. Notably, it achieves state-of-the-art performance on eight datasets and remains highly competitive on the Pubmed dataset.
- GADAM exhibits good compatibility, surpassing most methods that struggle to deliver satisfactory performance on datasets that contain either injected or real anomalies. In contrast, GADAM excels at effectively addressing diverse anomalies, whether they exhibit synthetic or real patterns.
- GADAM exhibits impressive scalability, distinguishing itself from most baseline models that encounter OOM when dealing with large-scale OGB datasets, which further reveals the high computation efficiency of our method.

## 4.3 DETAILED ANALYSIS FOR TWO STAGES

The tow-stage framework of GADAM prompts a natural question: **to what extent do these two stages enhance the model?** First, we investigate the impact of isolating the message passing on the effectiveness of LIM in the fitst stage. We evaluate this by comparing two variants: GADAM$_{local}$

Table 2: ROC-AUC comparison of model variants on five benchmark datasets.(**First** Second)

| Method | Cora | Citeseer | Pubmed | ACM | BlogCatalog |
|---|---|---|---|---|---|
| CoLA | 0.8801 | 0.8891 | 0.9435 | 0.7783 | 0.7807 |
| $GADAM_{local}^{GCN}$ | 0.6480 | 0.7281 | 0.7841 | 0.6691 | 0.4752 |
| $GADAM_{local}$ | 0.9256 | 0.9141 | 0.9510 | 0.9199 | 0.7625 |
| GADAM | **0.9556** | **0.9415** | **0.9581** | **0.9603** | **0.8117** |

and $GADAM_{local}^{GCN}$. In $GADAM_{local}$, anomaly detection relies solely on $\boldsymbol{S}^{local}$ obtained in the first stage, while in $GADAM_{local}^{GCN}$, we utilize a GCN for node embedding instead of an MLP. Next, we assess the effectiveness of the proposed enhancements in §3.1 by comparing $GADAM_{local}$ and CoLA. Finally, we evaluate the impact of the second stage by comparing GADAM and $GADAM_{local}$.

The experimental results presented in Tab. 2 reveal several key findings: (1) Message passing exerts a considerable negative impact on LIM ($GADAM_{local}^{GCN}$ v.s $GADAM_{local}$), thereby validating the effectiveness of our decoupling strategy. (2) Our proposed enhancement strategies within the MLP-based LIM exhibits a notable improvement in effectiveness ($GADAM_{local}$ v.s CoLA). (3) The design of the second stage significantly enhances the model's performance ($GADAM_{local}$ v.s GADAM).

## 4.4 EFFICIENCY COMPARISON

To further compare the efficiency and scalability, we count the total running time and maximum GPU memory consumption during model execution. As shown in Fig. 3, GADAM excels in both runtime and memory overhead, leading to significantly better efficiency and scalability .

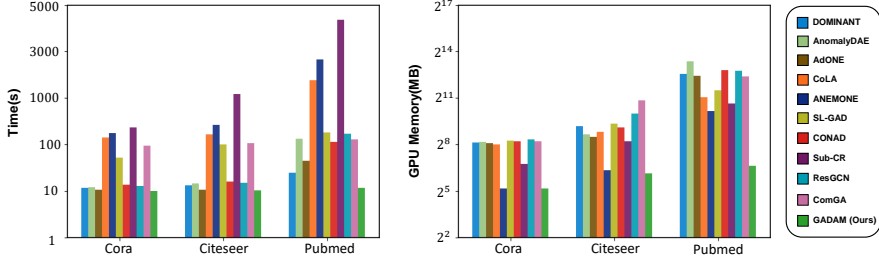

Figure 3: The runtime (left) and GPU overhead (right) of different methods. Noting that runtime includes data augmentation, model training, and anomaly score calculation procedures that may be necessary, and the maximum GPU memory requirement throughout the model's execution is shown.

## 4.5 ABLATION STUDIES

### 4.5.1 SIZE AND QUALITY OF HIGH-CONFIDENCE ANOMALY SET

In GADAM, two high-confidence sets are established to guide the second stage of learning. **An important question is that,what is the false rate of these pseudo labels and their effect on the training?** We present the analysis results in Fig. 4. Notably, given that normal nodes constitute the overwhelming majority, the high-confidence normal set exhibits high accuracy.Thus, the analysis about the high-confidence normal set is not reported.

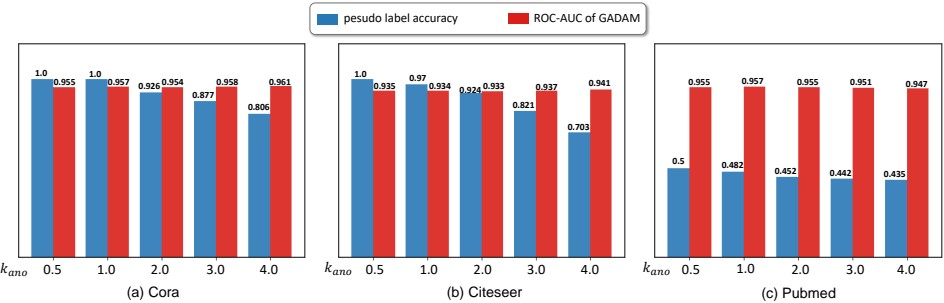

Figure 4: Pseudo label accuracy of high-confidence anomaly set and the model performance of GADAM with different $k_{ano}$ on three benchmark datasets.

Fig. 4 shows that the pseudo label accuracy tends to decrease as the size of the high-confidence abnormal set grows. However, GADAM maintains consistent performance across various set sizes. Additionally, our model maintains its effectiveness, even when the label accuracy of high-confidence anomaly set falls below 0.5 (e.g., the Pubmed dataset), showcasing remarkable robustness. An intuitive explanation for this phenomenon is that enlarging the set size exposes more anomalous signals to the model, which helps to prevent overfitting and mitigate the class imbalance.

### 4.5.2 ANALYSIS OF HYBRID ATTENTION MECHANISM

We present a set of ablation experiments to provide a more in-depth analysis of the hybrid attention mechanism. The results are shown in Tab. 3, revealing that a lack of (w/o attn) or blind message flow (fixed attn) is ineffective. Moreover, both pre-attention and post-attention contribute to model performance. Overall, the hybrid attention mechanism can achieve the best performance.

### 4.5.3 INFLUENCE OF EMBEDDING DIMENSION

Fig. 5 exhibits a rising trend in performance as the embedding dimension increases, followed by a stabilization phase. To strike a balance between effectiveness and computation efficiency, we set the embedding dimension to 64 for all datasets.

Table 3: Ablation study for hybrid attention. "w/o attn" denotes without message passing, "fixed attn" denotes the coefficient in Eq. (7) is set to a constant: $\alpha_i^{(t)} = \frac{1}{2}$.

| Variants | Cora | Citeseer | Pubmed |
|---|---|---|---|
| GADAM | **0.9556** | **0.9415** | **0.9581** |
| w/o attn | 0.9213 | 0.9117 | 0.9341 |
| fixed attn | 0.9275 | 0.9157 | 0.9441 |
| w/o pre-attn | 0.9492 | 0.9278 | 0.9504 |
| w/o post-attn | 0.9501 | 0.9301 | 0.9482 |

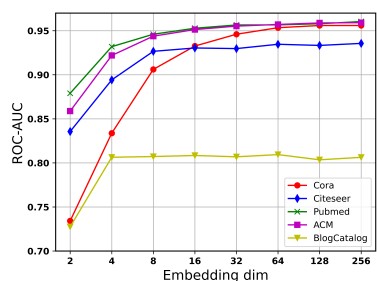

Figure 5: Influence of embedding dimension.

## 5 RELATED WORK

In this section, we present related works from two views: CL based methods and conflict-aware methods. CoLA (Liu et al., 2021) is the first to leverage CL for LIM. SL-GAD (Zheng et al., 2021) not only reconstructs the central node feature, but also performs contrastive learning between reconstructed and origin features. CONAD (Xu et al., 2022) designs four types of prior knowledge for abnormal nodes, and enables the model to learn prior knowledge to identify anomalies through CL. For conflict-aware methods, SpecAE (Li et al., 2019b) introduces a framework based on spectral convolution and deconvolution, utilizing Laplacian sharpening to magnify feature distances between anomalies and others. ResGCN (Pei et al., 2021) utilizes an MLP to model the attribute matrix, obtaining a residual matrix. Node residuals serve as both a measure of anomaly degree and the strength of message passing. ComGA (Luo et al., 2022) learns community-aware node representations to avoid excessive similarity of node characteristics across different communities. Also, there exist noteworthy works (Ding et al., 2021; Huang et al., 2023; Yang et al., 2023; Fathony et al., 2023; Wang et al., 2023) beyond the above views, providing a broader overview of UGAD.

## 6 CONCLUSION

In this paper, we investigated the shortcomings of existing unsupervised graph anomaly detection methods. We introduced a novel approach to perform local inconsistency mining and message passing decoupled, and further detect anomalies beyond the local perspective by incorporating adaptive message passing and global consistency discernment. The proposed GADAM, features a clear framework, high efficiency, and the ability to handle large-scale datasets. Extensive experimental results reveal the superiority of GADAM, and ablation studies provide a further understanding of GADAM. We hope our work can shed new insights on graph anomaly detection, and provide inspiration for real-world scenarios.

ACKNOWLEDGMENTS

This work was supported by the National Natural Science Foundation of China (#62102177), the Natural Science Foundation of Jiangsu Province (#BK20210181), the Key R&D Program of Jiangsu Province (#BE2021729), and the Collaborative Innovation Center of Novel Software Technology and Industrialization, Jiangsu, China.

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

# A    DATASET DETAILS

## A.1    ANOMALY INJECTION DETAILS

Anomalous nodes in seven real-world datasets are generated by the standard injection approach, including contextual anomaly and structural anomaly:

**Contextual Anomaly.** Contextual anomalies are nodes whose features are significantly different from neighboring nodes. $m$ nodes are chosen randomly to be regarded as contextual anomalies, while for each node $i$, $n$ random nodes are selected as candidates. Among the candidates, node $j$ is chosen with the largest Euclidean distance with node $i$: $||\boldsymbol{x_i} - \boldsymbol{x_j}||^2$, and we replace the feature $\boldsymbol{x_i}$ with $\boldsymbol{x_j}$. Therefore, $m$ nodes are injected as contextual anomalies. We follow previous works to set $n = 50$.

**Structural Anomaly.** Structural anomalies are nodes densely connected rather than sparsely connected regular nodes. We randomly select $m$ nodes and make them fully connected to form a clique, and all $m$ nodes are regarded as structural anomalies. The above process is repeated $n$ times, resulting in the injection of $m \times n$ structural anomalies. We follow previous works to set $m = 15$.

## A.2    ADDITIONAL DATASET INFORMATION

For datasets with injected anomalies, we keep two types of anomalies occupy the same proportion. For five benchmark datasets Cora, Citeseer, Pubmed, ACM, and BlogCatalog, we inject the same number of anomalies as the previous works. For two large datasets, we inject no more than 5% of anomalies. More details are shown in Tab. 4:

Table 4: Statistics of the datasets.

| Dataset | #Nodes | #Edges | Degree | #Anomalies | Ratio |
|---|---|---|---|---|---|
| Cora | 2,708 | 5,429 | 2.0 | 150 | 5.5% |
| Citeseer | 3,327 | 4,732 | 1.4 | 150 | 4.5% |
| Pubmed | 19,717 | 44,338 | 2.3 | 600 | 3.1% |
| ACM | 16,484 | 71,980 | 4.4 | 600 | 3.6% |
| BlogCatalog | 5,196 | 171,743 | 33.1 | 300 | 5.7% |
| ogbn-Arxiv | 169,343 | 1,166,243 | 6.9 | 6000 | 3.5% |
| ogbn-Products | 2,449,029 | 61,859,140 | 25.3 | 90000 | 3.6% |
| Books | 1,418 | 3,695 | 2.6 | 28 | 2.0% |
| Reddit | 10,984 | 168,016 | 15.3 | 366 | 3.3% |

**Cora**, **Citeseer**, **Pubmed**, and **ACM** are widely used citation networks. In these networks, nodes denote papers and edges denote citations between them. The feature of a node is a sparse bag-of-words vector extracted from a paper.

**BlogCatalog** is a social network platform that allows users to publicly share their blogs. The network structure is defined by the follower-followee relations. The node attributes consist of a list of tags that describe the users and their interests.

**ogbn-Arxiv** is a large-scale academic paper citation network, where nodes represent papers and edges represent the citations between them. Each node comes with a 128-dimensional feature vector obtained by averaging the embeddings of words in its title and abstract. The embeddings of individual words are computed by running the skip-gram model over the MAG corpus.

**ogbn-Products** is an Amazon product co-purchasing network, where nodes represent products sold in Amazon, and edges between two products indicate that the products are purchased together. Node features are generated by extracting bag-of-words features from the product descriptions followed by a Principal Component Analysis (PCA) to reduce the dimension to 100.

**Books** is a co-purchase network from Amazon. The attributes of nodes include prices, ratings, number of reviews, etc. The ground truth labels are derived from amazonfail tag information.

**Reddit** is a user-subreddit graph from the social media platform Reddit. The 1,000 most active subreddits and the 10,000 most active users are extracted as subreddit nodes and user nodes, respectively. Each user has a binary label indicating whether it has been banned by the platform, and the banned

users are treated as anomalies. The text of posts is converted into a feature, and the feature of users and subreddits are feature summation of posts they have.

# B ADDITIONAL EXPERIMENT SETTINGS AND DETAILS

## B.1 ENVIRONMENTS

Libraries that our method relies on are as follows: Python=3.7, CUDA=11.6, torch=1.12, dgl=0.9.0, networkx=2.6.3. For unsupervised baselines, we use benchmark implementations with pygod=0.3.0.

## B.2 HARDWARE

All experiments are performed on 1 Tesla V100 GPU with 32GB memory.

## B.3 DESCRIPTION OF BASELINES

- **DOMINANT** utilizes GCN as an encoder to extract embeddings. Then the adjacent matrix is reconstructed through the inner product of embeddings, and the feature matrix is reconstructed through another GCN.
- **AnomalyDAE** utilizes dual autoencoders with attention mechanisms to reconstruct the adjacent matrix and feature matrix.
- **AdONE** employs two autoencoders to reconstruct the adjacent matrix and feature matrix respectively. The effect of abnormal nodes on other nodes is minimized in the process of encoding in a decoupled way.
- **CoLA** adopts contrastive learning and uses GCN for node embedding. Two subgraphs are sampled via random walking for each node to form contrastive instances. The anomaly score is obtained based on the similarity difference between the central node and the two subgraphs.
- **ANEMONE** proposes a multi-scale contrastive approach based on CoLA, which further improves the detection performance.
- **SL-GAD** and CoLA share a similar contrast structure, but with a distinctive feature in SL-GAD. In SL-GAD, two positive instance subgraphs and two negative instance subgraphs are sampled for each node, creating two contrast instance pairs. This approach results in the formation of a multi-view contrast pattern. Furthermore, SL-GAD employs the positive instance subgraphs for reconstructing the center node, incorporating the reconstruction error into its anomaly score computation.
- **CONAD** proposes four anomaly types as prior human knowledge, and injects them into the graph with data augmentation. Contrastive learning is employed to guide the model to generate different representations for anomalies, and the anomaly score is obtained through reconstruction error.
- **Sub-CR** employs graph diffusion as structure enhancement, samples subgraphs from both the original graph and augmented graph and treats them as contrastive instances in local and global perspectives. Then multi-view contrastive learning and reconstruction error are used to generate the final anomaly score.
- **ResGCN** employs GCN for node representation learning while simultaneously reconstructing attribute and adjacency matrices. In addition to this, it employs an MLP to model the attribute matrix, resulting in a residual matrix. This approach is grounded in the intuition that larger residuals indicate a higher likelihood of a node being an anomaly. The residual values are harnessed as attention coefficients, regulating the information flow within and around the node in the GCN. Finally, the cumulative residual value is utilized to compute the anomaly score.
- **ComGA** identifies structural anomalies as nodes located at the edges of different communities that are connected to each other, which diverges from the current consensus. It focuses on making nodes within various communities distinguishable from one another, thus avoiding 'global smoothness'. To accomplish this, ComGA employs an autoencoder to model

the modular matrix and integrates it with GCN to enable nodes to acquire community-aware features. The anomaly scores are derived from the combined reconstruction errors of the adjacency matrix, identity matrix, and modular matrix.

## B.4 ADDITIONAL DESCRIPTION OF RELATED WORKS

In this section we provide a complementary introduction to methods that are not described in detail in related work:

- **AEGIS** (Ding et al., 2021) mainly focuses on inductive learning, including an anomaly-aware GNN layer and gives the model the ability to detect new anomalies by generative adversarial learning.

- **VGOD** (Huang et al., 2023) is designed specifically for two types of anomalies. It uses autoencoder to reconstruct the nodes and the reconstruction error is used to detect the contextual anomaly. At the same time, it uses the variance of the node's neighborhood features as a measure of the structural anomaly score, and finally the two anomaly scores are jointly used as the criteria for anomaly detection.

- **AHEAD** (Yang et al., 2023) is designed for anomaly detection in heterogeneous graphs. Three types of attention are designed for the encoder, and three types of reconstruction are used after the decoder. This design aims to achieve heterogeneity-aware unsupervised graph anomaly detection.

- **GraphBEAN** (Fathony et al., 2023) is designed for bipartite node-and-edge-attributed graphs, and detects anomalous nodes and edges simultaneously. Taking the example of consumer-purchase-product graph, GraphBEAN is designed to detect edges representing anomalous transactions, and users with anomalous behavior.

- **ACT** (Wang et al., 2023) focuses on cross-domain graph anomaly detection (CD-GAD), which describes the problem of detecting anomalous nodes in an unlabeled target graph using auxiliary, related source graphs with label nodes.

## B.5 EVALUATION METRICS

**ROC-AUC.** AUC computes the Area Under the Receiver Operating Characteristic Curve from predicted anomaly scores. The ROC curve is created by plotting the true positive rate against the false negative rate at various thresholds. In this paper, we regard anomalies as positive samples. AUC closer to 1 means that the model is more capable of recognizing anomalies.

**Average Precision.** AP provides insights into the precision of anomaly detection at various decision thresholds. It calculates the area under the Precision-Recall curve, which balances the effects of precision and recall. And a higher AP indicates a lower false-positive rate and false-negative rate.

**Recall@k.** Recall@k assesses the model's ability to identify anomalies among the top k ranked instances. It measures the proportion of true anomalies captured within the selected k instances, where we set k as the number of anomalies in each dataset. A higher Recall@k score signifies the detector's efficacy in giving priority to outlier detection over normal samples.

## B.6 MORE HYPERPARAMETER SETTINGS

The proposed model consists of two training stages: local inconsistency mining and global consistency discernment. The layer of MLP in each module is set to 1. The number of epochs, batch size, and learning rate for these two modules on each dataset are shown in Tab. 5.

## C ADDITIONAL EXPERIMENTAL RESULTS

In this section we provide experimental results under the Average Precision and Recall@k metrics as shown in Tab. 6 and Tab. 7 respectively.

Table 5: More details of hyperparameter settings, 0 denotes full batch. **Local** denotes local inconsistency mining and **Global** denotes global consistency discernment.

| Dataset | Local | | Global | | Batch size |
|---|---|---|---|---|---|
| | epoch | lr | epoch | lr | |
| Cora | 100 | 1e-3 | 50 | 5e-4 | 0 |
| Citeseer | 100 | 1e-3 | 50 | 5e-4 | 0 |
| Pubmed | 300 | 1e-3 | 200 | 5e-4 | 0 |
| ACM | 100 | 1e-3 | 30 | 5e-4 | 0 |
| BlogCatalog | 2000 | 2e-3 | 30 | 4e-4 | 0 |
| ogbn-Arxiv | 2000 | 2e-3 | 1000 | 4e-4 | 0 |
| ogbn-Products | 10 | 1e-4 | 10 | 3e-4 | 512 |
| Books | 100 | 1e-3 | 5 | 1e-5 | 0 |
| Reddit | 100 | 5e-4 | 50 | 5e-4 | 0 |

Table 6: Average precision comparison on nine benchmark datasets.(**First** Second) OOM: out of memory even if the batch size is set to 32.

| Method | Cora | Citeseer | Pubmed | ACM | BlogCatalog | ogbn-Arxiv | ogbn-Products | Books | Reddit |
|---|---|---|---|---|---|---|---|---|---|
| DOMINANT | 0.2010 | 0.2106 | 0.3176 | 0.1774 | 0.1519 | OOM | OOM | 0.019 | 0.037 |
| AnomalyDAE | 0.2831 | 0.2464 | 0.3037 | 0.2626 | 0.1658 | 0.1920 | OOM | 0.0194 | 0.040 |
| AdONE | 0.2331 | 0.3065 | 0.3733 | 0.2638 | 0.1811 | OOM | OOM | 0.0202 | 0.0320 |
| CoLA | 0.4700 | 0.3846 | 0.435 | 0.3465 | 0.1964 | **0.2056** | OOM | 0.0023 | 0.0437 |
| ANEMONE | 0.4483 | 0.4211 | 0.4644 | 0.3399 | 0.1804 | OOM | OOM | 0.0072 | 0.0415 |
| SL-GAD | 0.5232 | 0.4383 | 0.4861 | 0.3915 | 0.2683 | OOM | OOM | 0.0123 | 0.0330 |
| CONAD | 0.2101 | 0.3065 | 0.4038 | 0.3612 | 0.2132 | OOM | OOM | 0.0192 | 0.0326 |
| Sub-CR | 0.6240 | 0.4867 | **0.5413** | 0.4310 | 0.2438 | OOM | OOM | 0.0213 | 0.0463 |
| ResGCN | 0.4469 | 0.6446 | 0.3648 | 0.3804 | 0.2205 | OOM | OOM | 0.0179 | 0.0396 |
| ComGA | 0.5799 | 0.5823 | 0.5247 | 0.4128 | 0.2579 | OOM | OOM | 0.0259 | 0.0461 |
| GADAM | **0.7280** | **0.7512** | 0.4264 | **0.4446** | **0.2960** | 0.1948 | **0.2469** | **0.0279** | **0.0481** |

# D  DETAILED TIME COMPLEXITY ANALYSIS

In this section we analyze in detail the time complexity of the baselines, and present the results in Tab. 8 for easy comparison. The number of layers of the encoder and the dimension of node embeddings are considered constant for simplicity. $\mathcal{E}$ and $N$ denote the number of edges and the number of nodes in $\mathcal{G}$. $\kappa$ is the number of sampling neighbors for each node, $R$ is the number of evaluation rounds, $c$ is the number of nodes within the local subgraph, $\omega$ is the average node degree of $\mathcal{G}$, and $n$ is the number of nodes for data augmentation.

## D.1  DOMINANT

The time complexity of DOMINANT is mainly focused on (1) employing GCN for representation learning for all nodes, which is $\mathcal{O}(N\omega) = \mathcal{O}(\mathcal{E})$, (2) reconstructing the adjacency matrix takes $\mathcal{O}(N^2)$, and (3) reconstructing the feature matrix through another GCN, which is also $\mathcal{O}(\mathcal{E})$. Overall, the time complexity of DOMINANT is $\mathcal{O}(\mathcal{E} + N^2)$.

## D.2  ANOMALYDAE

AnomalyDAE has almost the same framework as DOMINANT, but utilizes a graph attention network (GAT) for representation learning. So the time complexity of AnomalyDAE is also $\mathcal{O}(\mathcal{E}+N^2)$.

Table 7: Recall@k comparison on nine benchmark datasets.(**First** Second) OOM: out of memory even if the batch size is set to 32.

| Method | Cora | Citeseer | Pubmed | ACM | BlogCatalog | ogbn-Arxiv | ogbn-Products | Books | Reddit |
|---|---|---|---|---|---|---|---|---|---|
| DOMINANT | 0.2383 | 0.2306 | 0.3476 | 0.1500 | 0.1998 | OOM | OOM | 0.0096 | 0.0095 |
| AnomalyDAE | 0.4451 | 0.3315 | 0.3885 | 0.2634 | 0.3162 | 0.1920 | OOM | 0.0116 | 0.0402 |
| AdONE | 0.3499 | 0.2969 | 0.3903 | 0.2543 | 0.2039 | OOM | OOM | **0.0239** | 0.0282 |
| CoLA | 0.4533 | 0.4067 | **0.4718** | 0.3478 | 0.2883 | 0.1997 | OOM | 0.0003 | 0.0463 |
| ANEMONE | 0.4768 | 0.4132 | 0.4560 | 0.3915 | 0.3029 | OOM | OOM | 0.0129 | 0.0534 |
| SL-GAD | 0.6155 | 0.4855 | 0.4251 | 0.3810 | 0.3274 | OOM | OOM | 0.0125 | 0.0473 |
| CONAD | 0.4253 | 0.3217 | 0.3699 | 0.3109 | 0.2229 | OOM | OOM | 0.0054 | 0.0177 |
| Sub-CR | 0.6684 | 0.5167 | 0.4713 | 0.4452 | 0.3142 | OOM | OOM | 0.0145 | 0.0553 |
| ResGCN | 0.5406 | 0.4855 | 0.3699 | 0.3810 | 0.2653 | OOM | OOM | 0.0072 | 0.0577 |
| ComGA | 0.6365 | 0.5576 | 0.3748 | 0.4334 | 0.2928 | OOM | OOM | 0.0094 | 0.0545 |
| GADAM | **0.7299** | **0.7120** | 0.4620 | **0.4590** | **0.3667** | **0.3212** | **0.2553** | 0.0143 | **0.0699** |

Table 8: Comparison of time complexity.

| Method | Overall Complexity | Preprocessing/Augmentation | Contrastive Learning | Reconstruction |
|---|---|---|---|---|
| DOMINANT | $\mathcal{O}(\mathcal{E} + N^2)$ | - | - | $\mathcal{O}(\mathcal{E} + N^2)$ |
| AnomalyDAE | $\mathcal{O}(\mathcal{E} + N^2)$ | - | - | $\mathcal{O}(\mathcal{E} + N^2)$ |
| AdONE | $\mathcal{O}(N\kappa)$ | $\mathcal{O}(N)$ | - | $\mathcal{O}(N\kappa)$ |
| CoLA | $\mathcal{O}(cNR(c + \omega))$ | $\mathcal{O}(cNR\omega)$ | $\mathcal{O}(c^2NR)$ | - |
| ANEMONE | $\mathcal{O}(cNR(c + \omega))$ | $\mathcal{O}(cNR\omega)$ | $\mathcal{O}(c^2NR)$ | - |
| SL-GAD | $\mathcal{O}(cNR(c + \omega))$ | $\mathcal{O}(cNR\omega)$ | $\mathcal{O}(c^2NR)$ | - |
| CONAD | $\mathcal{O}(n\omega + \mathcal{E} + N^2)$ | $\mathcal{O}(n\omega)$ | $\mathcal{O}(\mathcal{E})$ | $\mathcal{O}(\mathcal{E} + N^2)$ |
| Sub-CR | $\mathcal{O}(N + cN(c + \omega))$ | $\mathcal{O}(N + cN\omega)$ | $\mathcal{O}(c^2N)$ | $\mathcal{O}(c^2N)$ |
| ResGCN | $\mathcal{O}(\mathcal{E} + N^2)$ | - | - | $\mathcal{O}(\mathcal{E} + N^2)$ |
| ComGA | $\mathcal{O}(\mathcal{E} + N^2)$ | $\mathcal{O}(N^2)$ | - | $\mathcal{O}(\mathcal{E} + N^2)$ |
| GADAM (ours) | $\mathcal{O}(\mathcal{E} + N)$ | - | $\mathcal{O}(\mathcal{E} + N)$ | - |

## D.3 AdONE

(1) AdONE first employs random walk with restart to get an enhanced graph structure:

$$\hat{A} = \frac{1}{T}\sum_{t=1}^{T} P^t$$
$$P^t = rP^{t-1}D^{-1}A + (1 - r)P^0$$

where $P^t \in R^{N \times N}$ is the transition matrix, $D$ is the degree matrix, and $A$ is the adjacent matrix of $\mathcal{G}$. In practice, $T$ is a small constant, and the computation can be performed by a sparse operator in PyTorch as $D$ is a diagonal matrix. So the time complexity is $\mathcal{O}(N)$.

(2) AdONE employs two MLP-based autoencoders to reconstruct the adjacent matrix and feature matrix respectively, the time complexity is $\mathcal{O}(N)$.

(3) AdONE also samples $\kappa$ neighbors for each node, and measures the homophily of nodes using the difference between node and neighbor features. The time complexity for a node is $\mathcal{O}(\kappa)$, and the total time complexity becomes $\mathcal{O}(N\kappa)$. Overall, the time complexity of AdONE is $\mathcal{O}(N\kappa)$.

## D.4 CoLA

The time complexity of CoLA is mainly focused on three components:

(1) Instance pair sampling, the time complexity of each random walk with restart subgraph sampling is $\mathcal{O}(c\omega)$. In the inference phase, $R$ rounds of sampling for each node are performed, then the total time complexity becomes $\mathcal{O}(c\omega RN)$.

(2) GCN-based contrastive learning model, the time complexity is mainly generated by the GCN module, which is $\mathcal{O}(c^2)$ for each pair and $\mathcal{O}(c^2NR)$ for a total.

(3) For anomaly score computation, the time complexity is far less than the above two phases, so here we ignore this term. To sum up, the overall time complexity of CoLA is $\mathcal{O}(cNR(c + \omega))$.

## D.5 ANEMONE

ANEMONE has the same framework as CoLA, with an additional contrastive perspective. So the time complexity of ANEMONE is also $\mathcal{O}(cNR(c + \omega))$.

## D.6 SL-GAD

SL-GAD and CoLA share the same contrast learning model, but also use adjacency subgraphs to generate a representation of the central node and reconstruct it. So the time complexity of SL-GAD is also $\mathcal{O}(cNR(c + \omega))$.

## D.7 CONAD

The time complexity of CONAD is mainly focused on three components: (1) Data augmentation module, which selects $n$ nodes and performs four types of augmentation approaches as shown in Tab 9. The time complexity of 'Dense' and 'Outlying' is $\mathcal{O}(n\omega)$, and $\mathcal{O}(n)$ for 'Deviated' and 'Disproportionate'. So the time complexity ofthe data augmentation module is $\mathcal{O}(n\omega)$. (2) Contrastive learning based on node representation learning through GAT, which is $\mathcal{O}(\mathcal{E})$. (3) Reconstructing the adjacency matrix takes $\mathcal{O}(N^2)$, and reconstructing the feature matrix through another GAT, which is $\mathcal{O}(\mathcal{E})$. So the time complexity of reconstruction is $\mathcal{O}(\mathcal{E} + N^2)$. To sum up, the overall time complexity of CONAD is $\mathcal{O}(n\omega + \mathcal{E} + N^2)$.

Table 9: Data augmentation approaches of CONAD.

| Anomaly types | Prior Knowledge | Augmentation Approach |
|---|---|---|
| Dense | Anomalies has unusually high degree | Randomly connect with other nodes |
| Outlying | Anomalies has unusually low degree | Drop most of their edges |
| Deviated | Anomalies has deviated attribute from its neighbors | Replace their features using the approach in A.1 |
| Disproportionate | Anomalies has unusually small or large attribute | Largely scale up or scale down their attributes |

## D.8 SUB-CR

(1) Sub-CR first employs graph diffusion as structure enhancement:

$$S = \alpha(I - (1 - \alpha)D^{-1/2}AD^{-1/2})^{-1}$$

where $A \in R^{N \times N}$ is the adjacent matrix of $\mathcal{G}$, $I$ is the identity matrix, and $D$ is the degree matrix. In practice, the computation can be performed by a sparse operator in PyTorch, as $I$ and $D$ are both diagonal matrices. So the time complexity is $\mathcal{O}(N)$.

(2) Sub-CR adopts the same strategy as CoLA for instance pair sampling but without R-round sampling in the inference stage, so the the time complexity is $\mathcal{O}(Nc\omega)$.

(3) Sub-CR employs GCN for contrastive learning, the time complexity is mainly generated by the GCN module, which is $\mathcal{O}(c^2N)$.

(4) Reconstructing the feature matrix through another GCN, which is also $\mathcal{O}(c^2N)$. To sum up, the overall time complexity of Sub-CR is $\mathcal{O}(N + c^2N + Nc\omega) = \mathcal{O}(N + cN(c + \omega))$.

## D.9 RESGCN

The time complexity of ResGCN can be broken down into two main components: (1) The time complexity associated with residual learning using an MLP for the adjacency matrix $A$ is denoted as $\mathcal{O}(\mathcal{E})$. (2) The time complexity related to node representation learning using GCN, involving the reconstruction of adjacency subgraph, feature matrix, and adjacency matrix, is represented as $\mathcal{O}(\mathcal{E} + N^2)$. Thus the overall complexity is $\mathcal{O}(\mathcal{E} + N^2)$.

## D.10 COMGA

(1) In ComGA, the MLP with L layers serves as an autoencoder for both representation learning and the reconstruction of the modular matrix $\mathbf{B} \in \mathbb{R}^{N \times N}$. The time complexity for representation learning is $\mathcal{O}(N)$, with the number of MLP layers being disregarded. Meanwhile, the time complexity for reconstruction is $\mathcal{O}(N^2)$.

(2) Additionally, ComGA utilizes GCN for node feature learning and reconstructs both the adjacency and feature matrices, incurring a time complexity of $\mathcal{O}(\mathcal{E} + N^2)$.

Consequently, the overall complexity of ComGA is $\mathcal{O}(\mathcal{E} + N^2)$.

### D.11 THE PROPOSED GADAM

By ignoring the number of layers of encoder and the dimension of node embedding, the time complexity of GADAM simplifies to:

(1) For MLP-based LIM, the main time complexity is node representation learning which is $\mathcal{O}(N)$.

(2) For local-context aware message flow, the time complexity is mainly generated by node representation learning through MLP, which is $\mathcal{O}(N)$; and hybrid attention, which is $\mathcal{O}(N\omega)$ for pre-attention and $\mathcal{O}(N\omega)$ for post-attention. For anomaly score combination, the time complexity is ignored.

To sum up, the overall time complexity of GADAM is $\mathcal{O}(N\omega + N) = \mathcal{O}(\mathcal{E} + N)$.

## E VISUALIZATION OF ATTENTION

In this section, we present a visualization of the attention coefficients to provide an intuitive understanding of how attention evolves for different types of nodes during the training process. Fig. 6 illustrates the dynamics of attention across three benchmark datasets.

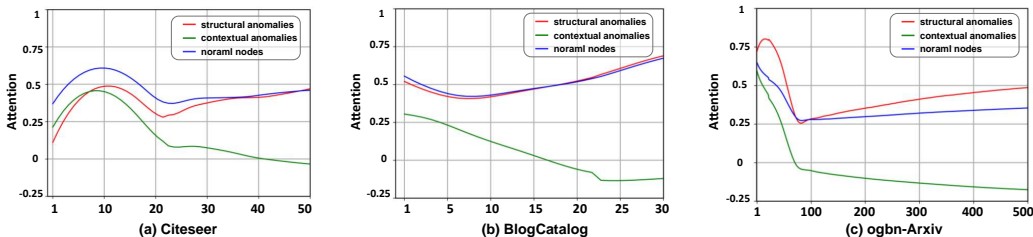

Figure 6: Changes in mean attention values for different node types across the training of the second stage. The X-axis represents the training epoch.

Based on the results depicted in Fig. 6, the following key observations can be made:

- The attention given to normal nodes and structural anomalies exhibits an increasing trend during training, whereas contextual anomalies displays a decreasing trend. This aligns with the motivations and objectives behind the hybrid attention mechanism.

- The attention given to normal nodes and structural anomalies far exceeds that allocated to contextual anomalies. This observation highlights the effectiveness of the hybrid attention mechanism in fine-grained control of node-specific message passing.

Another noteworthy point is that our attention is not strictly confined to the interval $[0, 1]$, as demonstrated by the fact that the attention of contextual anomalies drops below 0. This design is inspired by Li et al. (2019b). Formally, given the feature of a node $\boldsymbol{h}$ and the attention coefficient $\alpha$, let $\boldsymbol{e}_p$ be the embedding of its adjacent subgraph. Then, the adaptive message passing is performed as:

$$\boldsymbol{h} = \alpha \cdot \boldsymbol{e}_p + (1 - \alpha) \cdot \boldsymbol{h}$$

When the attention coefficient $\alpha > 0$, it signifies that the node integrates information from its surroundings, akin to a smoothing operation. Conversely, when the attention coefficient $\alpha < 0$, it suggests that the node is emphasizing its own features with greater weight, akin to a sharpening operation.

# F  THEORETICAL GUARANTEE OF GADAM

Inspired by Chen et al. (2022), we provide theoretical guarantee for the second stage of GADAM in this section. We begin with the formal problem statement. Given a graph $\mathcal{G}$ containing $N$ nodes, let $h_i$ be the embedding of node $v_i$, and let $q$ be the embedding for global context. Let $y_i$ be the label of node $v_i$, where $y_i = 0$ corresponds to normal nodes, and $y_i = 1$ indicates abnormal nodes. Assume there are $n$ normal nodes and $m$ abnormal nodes, satisfying $n + m = N$. We consider a relatively simple case by optimizing the loss function defined by Eq (13), where $n$ normal nodes are pulled closer to $q$, and the opposite for $m$ abnormal nodes:

$$\mathcal{L} = -\mathbb{E}_{i,j}[\log(q^T h_i) + \log(1 - q^T h_j)], \ \ where \ y_i = 0, \ y_j = 1$$

For better clarity, we use $h^+$ to represent the embedding of a normal node, and $h^-$ to represent the embedding of an abnormal node. We perform mean pooling on all nodes to obtain $q$, thus $q = \frac{1}{N}(\sum_{i=1}^{n} h_i^+ + \sum_{j=1}^{m} h_j^-)$. Then we have:

**Theorem 1**: Let $X^+$ denote the random variable of normal node and $p^+$ denote its marginal distribution, thus $X^+ \sim p^+(x^+)$. Likewise, we denote the abnormal node by $X^-$ and its marginal distribution by $p^-$. Assume $p^+$ and $p^-$ are mutually independent. Then we have: minimizing the loss $\mathcal{L}$ forms the lower bound of 1) the cross-entropy of the two data distributions $p^+$ and $p^-$ plus 2) the entropy of $p^-$. Formally,

$$\min \mathcal{L} \iff \max \left[ H(p^-, p^+) + H(p^-) \right] \tag{14}$$

**Proof of Theorem 1**:

$$\mathcal{L} = - \left( \underbrace{\mathbb{E}_{x^+}[\log(q^T h^+)]}_{first} + \underbrace{\mathbb{E}_{x^-}[\log(1 - q^T h^-)]}_{second} \right) \tag{15}$$

The first term in Eq.(15) pulls normal nodes and global context closer, and the second them pushes abnormal nodes and global context away. Then for the first term, we have:

$$q^T h^+ = \frac{1}{N}[\sum_{i=1}^{n}(h_i^+)^T h^+ + \sum_{j=1}^{m}(h_j^-)^T h^+]$$

Note that $n >> m$ and the similarity between normal nodes is predominant and usually large, so the term $q^T h^+$ is naturally large. Asymptotically, suppose we have $q^T h^+ = 1$, then minimizing $\mathcal{L}$ is equivalent to maximizing the second term, and we have:

$$\mathbb{E}_{x^-}[\log(1 - q^T h^-)] \leq \log \mathbb{E}_{x^-}[1 - q^T h^-]$$
$$= \log(-\mathbb{E}_{x^-}[q^T h^-])$$

The first inequality follows the Jensen Inequality based on the concavity of the log function, namely $\log(\mathbb{E}[x]) \geq \mathbb{E}[\log(x)]$. Therefore we consider the upper bound of the second term:

$$\max \ \mathbb{E}_{x^-}[\log(1 - q^T h^-)] \leq \max \ \log(-\mathbb{E}_{x^-}[q^T h^-])$$

We omit the log and convert the negative sign to get:

$$\max \ \log\left(-\mathbb{E}_{x^-}[q^T h^-]\right) \iff \min \ \mathbb{E}_{x^-}[q^T h^-]$$

$$= \min \ \mathbb{E}_{x^-} \frac{1}{N} \left[ \sum_{i=1}^{n}(h_i^+)^T h^- + \sum_{j=1}^{m}(h_j^-)^T h^-) \right]$$

$$= \frac{1}{N} \cdot \frac{1}{m} \sum_{j} \left[ \min\left(\sum_{i=1}^{n}(h_i^+)^T h_j^-\right) + \min\left(\sum_{k=1}^{m}(h_k^-)^T h_j^-\right) \right]$$

$$= \frac{1}{m} \sum_{j} [\min\left( \log\frac{1}{n} \sum_{i=1}^{n} e^{(h_i^+)^T h_j^-} + \log n \right)$$

$$+ \min\left( \log\frac{1}{m} \sum_{k=1}^{m} e^{(h_k^-)^T h_j^-} + \log m \right)]$$

$$= \min \ [-H(p^-, p^+) - H(p^-) + \log Z_{vMF}]$$

Inspired by the entropy estimation operation in Wang & Isola (2020), the last equality can be viewed as a resubstitution entropy estimator of $h^{+/-}$ (Lin, 1976) via a von Mises-Fisher (vMF) kernel density estimation (KDE), where $\log Z_{vMF}$ is the normalization constant.

# G  DETECTION PERFORMANCE ON STRUCTURAL AND CONTEXTUAL ANOMALIES

In this section, we study the performance of different algorithms on two types of injected anomalies, along with nodes that exhibit a dual type by manifesting as both contextual and structural anomalies (referred to as combined anomalies).

## G.1  PERFORMANCE ON STRUCTURAL AND CONTEXTUAL ANOMALIES

Three benchmark datasets were employed in our experimentation: **Cora** serves as a small, regular dataset with a low average node degree. **ACM**, on the other hand, represents a larger dataset with a substantial number of anomalies. Finally, **BlogCatalog** is characterized by a large average node degree, presenting a greater challenge for the detection of structural anomalies. We also selected four representative methods in the baseline families described in Sec. 4.1 to form a comparison experiment: DOMINANT, CoLA, Sub-CR and ComGA. Performance under AUC-ROC metirc is repoted in Tab. 10.

Table 10: ROC-AUC comparison on two types of anomalies.(**First** Second)

| Method | Cora | | ACM | | BlogCatalog | |
|---|---|---|---|---|---|---|
| | Contextual | Structural | Contextual | Structural | Contextual | Structural |
| DOMINANT | 0.7173 | 0.9312 | 0.7045 | 0.8182 | 0.7371 | 0.7664 |
| CoLA | 0.8934 | 0.8213 | 0.9103 | 0.8325 | 0.8585 | 0.7325 |
| Sub-CR | 0.9057 | 0.9293 | 0.7067 | 0.7913 | 0.8267 | 0.7573 |
| ComGA | 0.8173 | 0.9052 | 0.7525 | 0.8989 | 0.8294 | 0.7679 |
| GADAM | **0.9416** | **0.9929** | **0.9586** | **0.9721** | **0.8499** | **0.9094** |

The results show that:

- Most of the baselines struggles to balance the detection of both types of anomalies, while GADAM achieves the best.

- On the two sparser datasets, Cora and ACM, GADAM significantly outperforms baselines in structural anomaly detection. Even on the denser BlogCatalog dataset, GADAM continues to outperform all baselines, providing further validation of the effectiveness for structural anomaly detection.

Table 11: ROC-AUC comparison on combined anomalies.(**First** Second)

| Method | Cora $q = 15$ | Citeseer $q = 15$ | Pubmed $q = 60$ | ACM $q = 60$ | BlogCatalog $q = 30$ |
|---|---|---|---|---|---|
| DOMINANT | 0.8354 | 0.8498 | 0.7904 | 0.7448 | 0.7723 |
| CoLA | 0.8976 | 0.8928 | 0.9618 | 0.7909 | 0.7965 |
| Sub-CR | 0.9236 | 0.9400 | 0.9810 | 0.7518 | 0.7963 |
| ComGA | 0.8828 | 0.9335 | 0.9255 | 0.8433 | 0.8275 |
| GADAM | **0.9970** | **0.9994** | **0.9863** | **0.9884** | **0.9685** |

### G.2 PERFORMANCE ON COMBINED ANOMALIES

To further explore the performance of the model on combined anomalies , we generated a certain number of mixed anomalies on the three benchmark datasets. Specifically, we randomly select $q$ structural anomalies and employed the contextual anomaly injection method outlined in Appendix A.1 to substitute features for these nodes, thereby creating $q$ mixed anomalies. The detection performance on five benchmark datesets under the AUC-ROC metric is presented in Tab. 11.

The results show that GADAM has superior detection performance for combined anomalies. This success can be attributed to the first stage of GADAM captures well the contextual abnormal features exhibited by mixed anomalies. Moreover, in the second stage, GADAM facilitates the absorption of anomalous signals from the neighboring structural anomalies through adaptive message passing, further enhancing the detection of mixed anomalies.

