# OpenReview forum: "Boosting Graph Anomaly Detection with Adaptive Message Passing"
_ICLR.cc/2024/Conference — ICLR 2024 poster_

### Official Review · Reviewer_va1S · 2023-10-31

**Soundness:** 3 good
**Presentation:** 3 good
**Contribution:** 2 fair
**Rating:** 6
**Confidence:** 4

**Summary:**

In this paper, the authors address the challenge of unsupervised graph anomaly detection, a crucial task in various real-world applications. Existing methods often conflict in their focus between local inconsistency mining (LIM) and message passing. LIM emphasizes identifying high similarities between abnormal nodes and their neighbors, while message passing, commonly employed by graph neural networks (GNNs), tends to make connected nodes similar, leading to local anomaly signal loss. To reconcile this conflict, the authors propose GADAM, a novel framework that not only resolves the tension between LIM and message passing but also utilizes message passing to enhance anomaly detection through a unique approach to anomaly mining beyond LIM. The effectiveness and efficiency of GADAM are extensively evaluated on nine benchmark datasets, including two large-scale OGB datasets. The results demonstrate that GADAM outperforms existing state-of-the-art methods, showcasing superior effectiveness and efficiency in unsupervised graph anomaly detection.

**Strengths:**

- The proposed method is technically sound, i.e., to overcome the issues of LIM in previous anomaly detection methods.
- The proposed method has been tested on both synthetic and real-world anomalies and the experimental results show the effectiveness of the proposed method.
- The conducted experiments are comprehensive including performance comparison, efficiency comparison, and ablation studies.

**Weaknesses:**

- Although the effectiveness and efficiency of the proposed method have been empirically studied from different aspects, this is no theoretical guarantee of the proposed method.
- The first step of MLP-based LIM aims to identify local anomalies only. However, if the given data contains more contextual anomalies, will the proposed method fail to capture them?
The influence of different types of anomalies on the proposed method is not discussed.

**Questions:**

1. How to calculate H_{global} in details?
2. If there is no prior information about the ratio of anomalous nodes in a given graph, how should one select k_{ano}? If the selected k leads to more anomalies than actual ones, will the proposed method be impacted negatively?
3. What are the detailed steps in injecting anomalies? What the performance will change if different numbers of contextual or structural anomalies are injected (especially the proposed method first detects local structural anomalies)?

--------------------------

After rebuttal, my concerns have been addressed.

---

> ### Author Response · Authors · 2023-11-17
> **Response to reviewer va1S, Part I**
>
> We greatly appreciate your valuable time and constructive comments. We are pleased that you found our paper to be clear and of good quality. We hope our answers can fully address your concerns.
>
> **W1.** Although the effectiveness and efficiency of the proposed method have been empirically studied from different aspects, this is no theoretical guarantee of the proposed method.
>
> Inspired by CoLA[1], in the first stage of the model, we employs a "node vs. local subgraph" contrastive learning approach to extract local anomalies. In this process, the model aims to maximize the similarity between nodes and their adjacent subgraphs. Nodes that are challenging to optimize receive higher anomaly scores due to their high local inconsistency. The motivation for such a contrastive learning framework is intuitive and fits well with the goal of mining local inconsistency. Meanwhile, our contrast learning framework has demonstrated its effectiveness in local anomaly detection, as affirmed by a considerable body of work[1,2,3].
>
> In the second stage of the model, we focus on mining another kind of abnormal signals from the perspective of "node vs. global normal context". Inspired by previous work[4], we **theoretically demonstrate** that by optimizing the loss loss function Eq.(13), our model can make the distribution of learned abnormal node features spread uniformly as much as possible in the feature space. Thus GADAM can better deal with the diverse abnormal distributions.
>
> We begin with the formal problem statement. Given a graph $G$ containing $N$ nodes, let $h_i$ be the embedding of node $v_i$, and let $q$ be the embedding for global context. Let $y_i$ be the label of node $v_i$, where $y_i=0$ corresponds to normal nodes, and $y_i=1$ indicates abnormal nodes. Assume there are $n$ normal nodes and $m$ abnormal nodes, satisfying $n+m=N$. We consider a relatively simple case by optimizing the loss function defined by Eq (13), where $n$ normal nodes are pulled closer to $q$, and the opposite for $m$ abnormal nodes:
>
> $
> \mathcal{L} = -\mathbb{E}_{i,j}[\text{log}(q^Th_i) + \text{log}(1-q^Th_j)], where ~ y_i=0, y_j=1
> $
>
> For better clarity, we use $h^+$ to represent the embedding of a normal node, and $h^-$ to represent the embedding of an abnormal node. We perform mean pooling on all nodes to obtain $q$, thus $q = \frac{1}{N}(\sum_{i=1}^n h_i^+ + \sum_{j=1}^m h_j^-)$. Then we have:
>
> **Theorem 1**: Let $X^+$ denote the random variable of normal node and $p^+$ denote its marginal distribution, thus $X^+ \sim p^+(x^+)$. Likewise, we denote the abnormal node by $X^-$ and its marginal distribution by $p^-$. Assume $p^+$ and $p^-$ are mutually independent. Then we have: minimizing the loss $\mathcal{L}$ forms the lower bound of 1) the cross-entropy of the two data distributions $p^+$ and $p^-$ plus 2) the entropy of $p^-$. Formally, $\text{min}~ \mathcal{L} \iff \text{max}~[H(p^-,p^+) + H(p^-)]$.
>
> The **proof of Theorem 1** can be found in Appendix F in our revision. Below are some key conclusions drawn from Theorem 1:
> + The cross-entropy term $H(p^-,p^+)$ in Theorem1 links to the node-wise contrastive loss, which is proved to be the cross-entropy between the predicted labels and the ground truth labels[5]. This enables GADAM to distinguish between normal and abnormal nodes effectively.
> + Maximizing the entropy of abnormal nodes ($H(p^-)$) can be viewed as a form of data augmentation, as it enrich the information of minor classes, which helps to reduce the label imbalance issue. Also, it helps GADAM to better deal with the diverse abnormal distributions.
> + GADAM obtains global normal context through mean pooling on high-confidence normal set $V_n$. This approach is more representative of the distribution of normal nodes when compared to the global context $q$, as it involves less contamination.
>
> In summary,  as per Theorem 1, GADAM can effectively handle diverse distributions of anomalies and enhance its anomaly detection capabilities with a solid theoretical guarantee.
>
> **W2.** Thank you for your detailed review. We will address your conceens one by one:
>
> >(1) The first step of MLP-based LIM aims to identify local anomalies only.
>
> Actually, MLP-based LIM has the ability to detect some structural anomalies. It's worth noting that the neighbors of some structural anomalies may include normal nodes(e.g. node 7 in Figure 2 of our paper). Consequently, these structural anomalies can exhibit high local inconsistency, making them detected in the first stage. Hence, $S^{local}$ remains effective in detecting structural anomalies with this characteristic.
>
> >(2) If the given data contains more contextual anomalies, will the proposed method fail to capture them? The influence of different types of anomalies on the proposed method is not discussed.
>
> This is a good idea to do further testing of our model. We noticed that you asked a similar question in Q3, so we will answer it in detail in Q3.

---

> ### Author Response · Authors · 2023-11-17
> **Response to reviewer va1S, Part II**
>
> **Q1.** How to calculate H_{global} in details?
>
> Thanks for bringing this up! In the second stage, we deploy a new MLP on the raw node features and follow Eq.(3) to get $H_{global}$. After that we extract anomaly signals by considering the consistency between nodes and the global normal context. The learning process for $H_{global}$ is undertaken by optimizing Equation (13).
>
> **Q2.** If there is no prior information about the ratio of anomalous nodes in a given graph, how should one select k_{ano}? If the selected k leads to more anomalies than actual ones, will the proposed method be impacted negatively?
>
> Thanks for your thoughtful consideration. We'll respond to each in turn:
>
> >(1) How to select $k_{ano}$ if there is no prior information about the ratio of anomalous nodes?
>
> $k_{ano}$ determines the size of the high-confidence abnormal set $V_a$. So the setting of $k_{ano}$ relies on two principles: (1) making $V_a$ contain as many true anomalies as possible, and (2) making $V_a$ contain as few normal nodes as possible. Ideally, the optimal value of $k_{ano}$ should be the percentage of anomalies in each dataset, i.e., it relies on the percentage of anomalies in the dataset as a priori. But we kindly emphasize that **we do not rely on the anomaly proportion of the dataset as a priori when setting $k_{ano}$**. Given that anomalies typically represent minor classes, we just empirically set $k_{ano}$ within a reasonable interval, typically no greater than 5. Ablation studies conducted in Sec 4.5.1 also demonstrate the robustness of GADAM to different $k_{ano}$ values in this interval.
>
> >(2) If the selected k leads to more anomalies than actual ones, will the proposed method be impacted negatively?
>
> If $k_{ano}$ continues to increase, it will clearly have a negative impact on the model, as it will cause $V_a$ to be contaminated by more normal nodes, thus providing the wrong supervised signals for model training. Also, we did observe the phenomenon of negative impact on the Pubmed dataset from the ablation study of Sec 4.5.1, i.e., when $k_{ano}=4$ is larger than the actual percentage of anomalies in Pubmed (3.1%), there is a slight decrease in the AUC of the model. However, on other datasets, the $k_{ano}=5$ setting in our experiments leads to more anomalies than actual ones on many datasets according to Appendix A2, but we empirically find that the model still reaches state-of-the-art performance by a large advantage.
>
> In summary,we set $k_{ano}$ in a reasonable interval (not greater than 5) . Also within this interval GADAM shows strong robustness as shown in ablation studies conducted in Sec 4.5.1.
>
> **Q3.** Thank you for your insightful question! Below are the detailed responses to each question:
>
> >(1) What are the detailed steps in injecting anomalies ?
>
> In fact we provide a detailed description in Appendix A.1. For your convenience, here's a quick summary:
>
> To inject a contextual anomaly, we randomly choose a node $i$ and select $n$ nodes as candidates.  Among the candidates, we pick node $j$ based on the largest Euclidean distance from node $i$: $||x_i − x_j || ^2$, and we replace the feature $x_i$ with $x_j$ to generate contextual anomaly.
>
> For structural anomalies, we randomly select $m$ nodes and make them fully connected to form a cluster, and all $m$ nodes are regarded as structural anomalies.
>
> The aforementioned methods can be applied to generate a specified number of anomalies, as detailed in Appendix A.1.

---

> ### Author Response · Authors · 2023-11-17
> **Response to reviewer va1S, Part III**
>
> >(2) What the performance will change if different numbers of contextual or structural anomalies are injected ?
>
> Thanks for your valuable suggestion! To comprehensively assess the performance of our model with different numbers of anomalies injected, we selected three representative benchmark datasets for detailed testing: **Cora** represents a small dataset that is regular and has a low average node degree, **ACM** represents a larger dataset that contains a high number of anomalies, and **BlogCatalog** represents a dataset with a large average node degree that may pose a greater challenge to structural anomaly detection. We kept the total number of anomalous nodes unchanged (consistent with our paper, see appendix A.2) and conducted tests under three scenarios: contextual anomalies only, a balanced number of anomalies, and structural anomalies only.  We measured GADAM's effectiveness using the AUC-ROC metric, with bolded results indicating the scenarios in which GADAM excels in detecting the two types of anomalies. Using the Cora dataset as an example, the bolded results show that GADAM is best at detecting contextual anomalies in the contextual only case and structural anomalies in the structural only case:
>
> **Cora**
> | contextual only |             | balanced anomalies |              | structural only |               |
> | :-------------- | ----------- | ------------------ | ------------ | --------------- | ------------- |
> | contextual=150  | structual=0 | contextual=75      | structual=75 | contextual=0    | structual=150 |
> | **0.9425**      | \           | 0.9416             | 0.9929       | \               | **0.9962**    |
>
> **ACM**
>
> | contextual only |             | balanced anomalies |               | structural only |               |
> | --------------- | ----------- | ------------------ | ------------- | --------------- | ------------- |
> | contextual=600  | structual=0 | contextual=300     | structual=300 | contextual=0    | structual=600 |
> | **0.9607**      | \           | 0.9586             | **0.9721**    | \               | 0.9678        |
>
> **BlogCatalog**
>
> | contextual only |             | balanced anomalies |               | structural only |               |
> | --------------- | ----------- | ------------------ | ------------- | --------------- | ------------- |
> | contextual=300  | structual=0 | contextual=150     | structual=150 | contextual=0    | structual=300 |
> | **0.9333**      | \           | 0.9094             | **0.7763**    | \               | 0.7553        |
>
> The experimental results reveal the following insights:
>
> + Regarding the detection of contextual anomalies, our model consistently demonstrates optimal performance in scenarios where only contextual anomalies are included. This effectively addresses the concern expressed in W2 regarding the potential impact of continuously adding contextual anomalies. It serves as validation for the excellent performance of our model in mining local anomaly signals using the conflict-free Local Inconsistency Mining (LIM) approach.
> + In detecting structural anomalies, our model achieves competitive results when only structural anomalies are included, as compared to the scenario with a balanced number of anomalies. Surprisingly, it even outperforms on the Cora dataset. This results show that GADAM is also a strong detector of structural anomalies.
>
> In summary, our model is capable of handling extreme anomaly distributions (contextual/structural only), and achieves very good performance in the more common balanced case. We appreciate your insightful questions, and we will integrate them in the later revision, which will help us to further improve the paper!
>
> **Reference**
>
> [1] Anomaly detection on attributed networks via contrastive self-supervised learning (TNNLS 2021)
>
> [2] Anemone: graph anomaly detection with multi-scale contrastive learning (CIKM 2021)
>
> [3] Reconstruction Enhanced Multi-View Contrastive Learning for Anomaly Detection on Attributed Networks (IJCAI 2022)
>
> [4] Gccad: Graph contrastive coding for anomaly detection (TKDE 2022)
>
> [5] A unifying mutual information view of metric learning: cross-entropy vs. pairwise losses (ICCV 2020)

---

> > ### Comment · Reviewer_va1S · 2023-11-23
> > **Thanks for response**
> >
> > Thanks for the detailed response especially the detailed explanation in Q2. These answers addressed my concerns. Therefore, I will increase my rating.

---

> > > ### Author Response · Authors · 2023-11-23
> > > **Thanks for your comment**
> > >
> > > We are glad that the responses have addressed your concerns. Thanks for your valuable time and kind comment!

---

### Official Review · Reviewer_6vV6 · 2023-10-31

**Soundness:** 3 good
**Presentation:** 3 good
**Contribution:** 3 good
**Rating:** 6
**Confidence:** 3

**Summary:**

The authors propose a model for unsupervised anomaly detection in graphs. In particular, the model focuses on 2 types of anomalies: 1) contextual (a node having attributes that are very different from its neighbors') and 2) structural (dense subgraphs). The innovation is a two-stage model that first uses an MLP to find possible contextual anomalies and then uses a message-passing graph neural network with an attention mechanism to find dense subgraphs of anomalous nodes.

**Strengths:**

- Takes inspiration of previous work on contrastive learning and recent work on attention for message passing
- Competitive time complexity
- Experimental evaluation is fairly comprehensive and shows modest improvement over existing methods, of which there are plenty

**Weaknesses:**

- Performance reported for competing methods is lower than in the papers where these methods were proposed. It raises the question of whether the models are sufficiently tuned
- No comparison to methods in the vast literature outside deep learning / neural networks. See for example https://arxiv.org/abs/1404.4679

**Questions:**

- The AnomalyDAE paper reports an AUC of over 97 for the BlogCatalog dataset. Do you have an explanation for the discrepancy in your paper (AUC of 76.58)? Were the authors of that paper injecting anomalies in a totally different way?

- The AUC for Sub-CR, DOMINANT, and  COLA is higher in the Sub-CR paper. Do you think hyperparameters for the baseline models are sufficiently tuned?

---

> ### Author Response · Authors · 2023-11-17
> **Response to reviewer 6vV6, Part I**
>
> Thanks a lot for your insightful comments and detailed suggestions, which are very helpful for us to further improve this paper. We hope our following answers will address the points you have raised and improve your view of our work.
>
> **W1.** Performance reported for competing methods is lower than in the papers where these methods were proposed. It raises the question of whether the models are sufficiently tuned
>
> Thanks for your careful review and bringing this up! We will provide a more detailed explanation in our responses to Q1 and Q2 below. In summary, we adhered to the hyperparameters recommended in the original papers and employed a standardized dataset generated uniformly with PyGod[1] library to ensure fair comparisons.
>
> **W2.** No comparison to methods in the vast literature outside deep learning / neural networks. See for example https://arxiv.org/abs/1404.4679
>
> Thanks for pointing this out! After a thorough examination of the relevant literature, we have identified the following key findings:
>
> + We have incorporated the definition of anomaly from the literature into our paper. Specifically, in Section 2.1.2, it is mentioned that community based methods identify anomalies by finding densely connected nodes, which is consistent with our definition of structural anomaly.  Additionally, in Section 2.2.2, some methods mark nodes with deviate attributes from other members in a community as anomalies, which is consistent with our definition of contextual anomaly.  Consequently, certain methods from the literature hold relevance as baselines.
> + Many methods discussed in the literature operate under a singular assumption about anomalies, tailored for a specific anomaly type. For instance, LOF[2] relies solely on node attributes to identify contextual anomalies by gauging the degree of deviation of nodes and their neighbors. On the other hand, SCAN[3] relies solely on structural information within the graph, detecting clusters through clustering for the identification of structural anomalies. In contrast, existing deep learning methods go beyond these limitations by adeptly utilizing both graph attributes and structural information, enabling a more comprehensive anomaly detection approach.
>
> Indeed, some methods outside deep learning / neural networks continue to hold relevance in contemporary anomaly detection. We refer to the benchmark[1] and chose LOF and SCAN as representative methods. Additionally, we included the widely adopted isolation forest (IF)[4] for further evaluation. Experimental results on three benchmark datasets are as follows:
>
> AUC-ROC：
>
> |              | Cora       |            |            | ACM        |            |            | BlogCatalog |            |            |
> | ------------ | ---------- | ---------- | ---------- | ---------- | ---------- | ---------- | ----------- | ---------- | ---------- |
> |              | contextual | structural | overall    | contextual | structural | overall    | contextual  | structural | overall    |
> | LOF          | 0.862      | 0.5156     | 0.7012     | 0.6167     | 0.4543     | 0.5567     | 0.3738      | 0.4741     | 0.4146     |
> | SCAN         | 0.4548     | 0.8121     | 0.6463     | 0.4534     | 0.8182     | 0.6678     | 0.4970      | 0.7579     | 0.6564     |
> | IF           | 0.7798     | 0.5054     | 0.6019     | 0.5655     | 0.5443     | 0.5602     | 0.6481      | 0.5024     | 0.5917     |
> | GADAM (ours) | **0.9416** | **0.9929** | **0.9556** | **0.9586** | **0.9721** | **0.9603** | **0.9094**  | **0.7763** | **0.8117** |
>
> The experimental results indicate that LOF and SCAN struggle to balance performance in detecting both anomaly types, aligning with our own observations. Furthermore, IF doesn't exhibit effectiveness, likely due to its lack of specialization for graph-based data. In contrast, our proposed model showcases effective detection capabilities for both contextual and structural anomalies, leveraging the combined information from graph attributes and structure.
>
> We sincerely appreciate your insightful review, and we will integrate the experimental results into a later revision with references to these literatures. Thank you again for enriching our paper with your valuable comment!

---

> ### Author Response · Authors · 2023-11-17
> **Response to reviewer 6vV6, Part II**
>
> **Q1.** The AnomalyDAE paper reports an AUC of over 97 for the BlogCatalog dataset. Do you have an explanation for the discrepancy in your paper (AUC of 76.58)? Were the authors of that paper injecting anomalies in a totally different way?
>
> Thank you for bringing up this concern! We've thoroughly reviewed the original AnomalyDAE paper and its associated source code[9]. However, it appears that the anomaly injection method is not clearly outlined in the article, and the provided source code only includes the BlogCatalog dataset with anomaly labels. To ensure a fair and comprehensive comparison, we utilized PyGOD[1], an open library from the NIPS'22 benchmark for unsupervised graph anomaly detection. We applied this library uniformly across all datasets to generate anomalies, striving for consistency and fairness.
>
> Simultaneously, we conducted experiments using the parameter configurations specified in the original AnomalyDAE paper. The obtained results align with those reported in the updated work[6]. Regrettably, due to the lack of clarity on how AnomalyDAE injects anomalies and the absence of support for other datasets in its source code, we couldn't replicate experiments on additional datasets using the provided code. We made diligent efforts to ensure the fairness of our experiments through PyGOD, and observed performance variations in AnomalyDAE may be dataset-dependent.
>
> **Q2.** The AUC for Sub-CR, DOMINANT, and COLA is higher in the Sub-CR paper. Do you think hyperparameters for the baseline models are sufficiently tuned?
>
> Thank you for your thoughtful consideration. Regarding CoLA and DOMINANT, we employed PyGOD for their implementation and adhered to the parameters suggested in the original papers. We have carefully compared the results with those reported in past works[5, 6, 7] and believe that the results we obtained for these two models are in the normal range.
>
> Concerning Sub-CR, as it lacks open-source availability, we initiated communication with the authors to obtain the source code. Unfortunately, the provided source code only includes prepared datasets without an interface for anomaly injection. Consequently, we injected anomalies and tested model perfromance through PyGOD's unified interface to ensure a fair comparison. Additionally, we maintained consistency by employing the same hyperparameter settings as specified in the original paper[8].
>
> Additionally, it's important to note that the results of the experiments will also vary with the anomaly injection implementation, software library versions, and hardware environments. In our efforts to uphold fairness, we have standardized the processes of dataset generation, baseline implementation, and model evaluation through PyGOD. We appreciate your understanding and consideration in this regard.
>
> **Reference**
>
> [1] BOND: Benchmarking Unsupervised Outlier Node Detection on Static Attributed Graphs (NIPS 2022)
>
> [2] LOF: Identifying Density-Based Local Outliers (SIGMOD 2000)
>
> [3] Scan: a structural clustering algorithm for networks. (KDD 2007)
>
> [4] Isolation forest (ICDM 2008)
>
> [5] Anemone: graph anomaly detection with multi-scale contrastive learning (CIKM 2021)
>
> [6] ComGA: Community-Aware Attributed Graph Anomaly Detection (WSDM 2022)
>
> [7] Generative and contrastive self-supervised learning for graph anomaly detection (TKDE 2021)
>
> [8] Reconstruction Enhanced Multi-View Contrastive Learning for Anomaly Detection on Attributed Networks (IJCAI 2022)
>
> [9] https://github.com/haoyfan/AnomalyDAE

---

### Official Review · Reviewer_deYB · 2023-11-03

**Soundness:** 3 good
**Presentation:** 3 good
**Contribution:** 3 good
**Rating:** 6
**Confidence:** 4

**Summary:**

The authors investigate the two main paradigms, both used in many unsupervised graph anomaly detection models, message passings (MP) and local inconsistency mining (LIM). Traditionally, many graph anomaly detection models utilize the LIM for determining nodes that have inconsistency with their surroundings and mark them as anomalous. However, the use of MP with many GNN layers tends to smooth out everything, which decreases the contrastive difference between a node and its surroundings. The authors aim to address this conflict by proposing GADAM, a method that first performs LIM via a regular MLP network to get local anomaly scores without using message passing. It then combines these local scores with global scores that use an attention-based adaptive message passing that enables nodes to selectively absorb normal/abnormal signals from their surroundings. The authors then demonstrate the benefit of the models compared with baselines on several public datasets.

**Strengths:**

Strengths:
- An interesting approach in unsupervised anomaly detection. The over-smoothing problem is a well-known problem in GNN, and it may impact the performance of GNN-based anomaly detection models. Most unsupervised graph anomaly detection models work by finding inconsistent nodes among the neighbors. The use of message passing in GNN may impact the finding as the over-smoothing problem may make the nodes less inconsistent.
I would argue that the conflict between LIM and overs-smoothing could be resolved within a message-passing framework by designing a better message-passing flow, imposing stricter bottlenecks in the encoder-decoder process, or other means. However, the paper proposed a different way outside the message-passing framework.

- The methods combine both local perspective and global perspective to create combined perspective anomaly scores.
- The authors demonstrate the benefit of the model against many GNN baselines

**Weaknesses:**

I have a few concerns and questions regarding the paper:
1) In the contrastive pair construction (Section 3.1 item (2)), the authors mentioned that the method uses the complete adjacency subgraph. Does it mean that it includes all 1-hop neighbors? If that's the case, then the receptive field of the method is limited to just the immediate neighbors. This may limit the expressiveness of the model. There may be cases where determining normal/abnormal nodes requires more than just looking at the immediate neighbors.
2) If in (1), it is not just 1-hop neighbors but all k-hop neighbors, then the size of the subgraphs can be potentially very large in densely connected graphs. How does the model address that?
3) In the shuffling process, there are some chances that neighboring nodes are accidentally picked as negative pairs. How does the model account for those cases?
4) The representations of a subgraph G in the local context are just an average of the representation of the node embeddings in G. Then, the anomaly scores for each node are measured by the difference between its node representation and the subgraph representation. It is possible that the node embedding of each neighbor is really different than the target node's embedding, but the average of the neighbor embeddings is similar to the target node. Doesn't it also introduce an over-smoothing problem? It could be worse than the standard message passing, where the aggregation is not just standard averaging but parameterized by a weight matrix. Here, the aggregation is just a plain average.
5) The use of attention mechanisms may be counter-intuitive in unsupervised anomaly detection that is based on finding abnormal signals from the surroundings. The ability of the attention mechanism to ignore certain signals may make the model ignore suspicious signals, which will result in an abnormal case tagged as a normal case. Has the author tried to replace the attention mechanism with another mechanism, like convolution, that forces the model to utilize all signals?
6) The experiment results of the baselines presented in the paper are rather different from the ones reported in the original paper for the same datasets. For example, in the original AnomalyDAE paper, the AUC are 97.81 (BlogCatalog), 90.05 (ACM); whereas in this paper, they are 76.58 (BlogCatalog), 75/16 (ACM). Could the authors explain more about it?
7) How to decide topk% included in the second stage. How does the selection of this percentage affect the results?
8) The paper is missing some unsupervised graph anomaly detection papers:
     - Ding, Kaize, Jundong Li, Nitin Agarwal, and Huan Liu. "Inductive anomaly detection on attributed networks." In Proceedings of the twenty-ninth international conference on international joint conferences on artificial intelligence, pp. 1288-1294. 2021.
     - Huang, Yihong, Liping Wang, Fan Zhang, and Xuemin Lin. "Are we really making much progress in unsupervised graph outlier detection? Revisiting the problem with new insight and superior method." arXiv preprint arXiv:2210.12941 (2022).
     - Yang, Shujie, Binchi Zhang, Shangbin Feng, Zhanxuan Tan, Qinghua Zheng, Jun Zhou, and Minnan Luo. "Ahead: A triple attention based heterogeneous graph anomaly detection approach." In Chinese Intelligent Automation Conference, pp. 542-552. Singapore: Springer Nature Singapore, 2023.
     - Fathony, Rizal, Jenn Ng, and Jia Chen. "Interaction-Focused Anomaly Detection on Bipartite Node-and-Edge-Attributed Graphs." In 2023 International Joint Conference on Neural Networks (IJCNN), pp. 1-10. IEEE, 2023.
     - Wang, Qizhou, Guansong Pang, Mahsa Salehi, Wray Buntine, and Christopher Leckie. "Cross-domain graph anomaly detection via anomaly-aware contrastive alignment." In Proceedings of the AAAI Conference on Artificial Intelligence, vol. 37, no. 4, pp. 4676-4684. 2023.

**Questions:**

Please answer my questions in the previous section.

---

> ### Author Response · Authors · 2023-11-17
> **Response to reviewer deYB, Part I**
>
> We greatly appreciate your valuable time and constructive comments. We are pleased that you found our paper to be clear and of good quality. We hope our answers can fully address your concerns.
>
> **W1.** Thanks for your detailed review! We will address your concerns one by one:
> >(1)Does it mean that it includes all 1-hop neighbors?
>
> Yes, we construct positive contrastive instance using the adjacent subgraph containing all 1-hop neighbors.
>
> >(2)The expressiveness of the model may be limited, as there may be cases where determining normal/abnormal nodes requires more than just looking at 1-hop neighbors.
>
> We acknowledge the possibility you've pointed out. Our choice to use only 1-hop neighbors is based on several considerations:
>
> + **Contextual Anomaly Characteristics:** Considering the nature of contextual anomalies, their primary feature is the significant inconsistency with their immediate neighbors. Utilizing 1-hop neighbors, which provides the most direct local contextual information, proves highly effective in constructing contrastive learning samples to extract the local inconsistency of nodes.
>
> + **Normal Node Perspective:** Multi-hop neighbor information may encompass more abnormal nodes from the viewpoint of normal nodes, potentially contaminating their contextual information. For instance, node 11 in Fig.2 of our paper, employing 2-hop neighbors will introduce a substantial number of abnormal nodes, leading the model to inaccurately identify node 11 as an anomaly. Thus, while multi-hop neighbors could enhance the receptive field, they introduce uncontrollable factors into the node's context, potentially hindering model performance.
>
> + **Efficiency Considerations:** As you rightly highlighted in W2, utilizing multi-hop neighbors increases both computational and memory overhead for the model. Both the efficiency and scalability of the model are also crucial aspects that we consider.
>
> + **Experimental Results:** We have empirically observed that relying solely on 1-hop neighbor enables GADAM to outperform current baselines significantly. This indicates that our approach achieves a favorable balance between model effectiveness and computational efficiency.
>
> **W2.** If in (1), it is not just 1-hop neighbors but all k-hop neighbors, then the size of the subgraphs can be potentially very large in densely connected graphs. How does the model address that?
>
> As mentioned above, our model uses only 1-hop neighbors, so the size of the subgraphs will not be too large. However, there are works such as CoLA[1] and ANEMONE[2] that utilize random walks to sample multi-hop neighbors for nodes, aiming to enhance the model's receptive filed while reducing the size of the neighborhood subgraph.  Nevertheless, these approaches have significant drawbacks:
>
> + Neighbor information obtained through sampling introduces considerable randomness, as explicitly stated in the original paper[1]. To address this randomness, CoLA constructs multiple contrastive instances by sampling each node with $R$ rounds in the final computation of the anomaly score, as described in Sec 2.2 in our paper. In CoLA, $R$ is recommended to be set to 256, meaning each node needs to sample 256 neighborhood subgraphs for stable and competitive results, imposing a significant computational overhead.
> + As we mentioned in W1, utilizing multi-hop neighbors may contaminate a node's neighborhood information to a greater extent, a factor that has been overlooked in existing works.
>
> What's more, our experimental results show that our model not only surpasses them significantly in terms of computational efficiency and GPU memory overhead but also outperforms them in terms of detection performance. We hope our explanations have addressed your queries regarding neighborhood subgraph construction. Thank you for your question, which has provided us with the opportunity for further clarification!
>
> **W3.** In the shuffling process, there are some chances that neighboring nodes are accidentally picked as negative pairs. How does the model account for those cases?
>
> Thanks for your detailed review. We did take this into account in our code implementation (utils.py/idx_sample), and we utilized a pseudo-shuffle approach. Suppose `idx` is a node index vector of length N, where `idx = [0, 1, ..., N-1]`. We implemented the shuffle by first generating a random number `rand` in the range `[1, N)`, and the negative instance index corresponding to node `i` is obtained by `neg_idx[i] = (i + rand) % N`, where `% ` denotes the modulo. For example, with `N=5` and `idx = [0, 1, 2, 3, 4]`, if `rand = 2`, then `neg_idx = [2, 3, 4, 0, 1]`. This approach ensures that the negative instance of any node will not be itself.

---

> ### Author Response · Authors · 2023-11-17
> **Response to reviewer deYB, Part II**
>
> **W4.** The simple mean pooling method for subgraph embedding may introduce an over-smoothing problem. It could be worse than the standard message passing, where the aggregation is not just standard averaging but parameterized by a weight matrix.
>
> We acknowledge your concerns. However, we empirically found that using only 1-hop neighbors and employing a simple mean pooling method for subgraph embedding is effective. **Introducing more parameters to learn subgraph embedding may lead to potential overfitting issues**. To address your concern, we experimented with using GCN and GAT instead of mean pooling in the first stage for subgraph embedding, while keeping the rest of GADAM unchanged. The results under AUC-ROC metric are as follows:
>
> |              | Cora       | Citeseer   | Pubmed     | ACM        | BlogCatalog | Books      | Reddit     |
> | ------------ | ---------- | ---------- | ---------- | ---------- | ----------- | ---------- | ---------- |
> | GCN          | 0.6322     | 0.7995     | 0.7561     | 0.5803     | 0.5175      | 0.5328     | 0.4242     |
> | GAT          | 0.4727     | 0.6009     | 0.7867     | 0.4001     | 0.6657      | 0.534      | 0.387      |
> | Mean pooling | **0.9556** | **0.9415** | **0.9581** | **0.9603** | **0.8117**  | **0.5983** | **0.5809** |
>
> The experimental results demonstrate that using more advanced methods for subgraph embedding results in a significant performance drop. The primary reason for this is that the first stage of GADAM operates in an unsupervised manner, where the model enhances the consistency of all nodes and their neighbors by optimizing the loss function in Eq.(1). In this process, most nodes with high consistency are further optimized by the model, while nodes with lower consistency are identified as anomalies due to the optimization difficulty. Introducing more parameters increases the expressive power of the model, but it also leads to overfitting of the loss function, resulting in a loss of local inconsistency of the anomalous nodes.
>
>  **W5.** Issues about hybrid attention mechanism.
>
> Thanks for your thorough consideration! We will address your concerns one by one:
>
> >(1)The use of attention mechanisms may be counter-intuitive in unsupervised anomaly detection that is based on finding abnormal signals from the surroundings.
>
> Our attention mechanism doesn't solely rely on anomaly scores; we incorporate post-attention based on feature similarity, aligning with some traditional attention mechanisms (e.g., GAT). These two attentions are dynamically weighted to create a hybrid attention mechanism, as demonstrated to be highly effective in the ablation study in Sec 4.5.2.
>
> >(2) The ability of the attention mechanism to ignore certain signals may make the model ignore suspicious signals.
>
> If we interpret your concern correctly, you're highlighting that our attention mechanism operates at the "node-subgraph" level, while traditional attention mechanisms like GAT operate at the more fine-grained "node-node" level. To address your concerns, we conducted additional experiments. In the second stage, we replaced the MLP+hybrid attention mechanism with GCN and GAT to extract features for nodes, while keeping the rest of GADAM unchanged. The experimental results under AUC-ROC metric are as follows:
>
> |                 | Cora       | Citeseer   | Pubmed     | ACM        | BlogCatalog | Books      | Reddit     |
> | --------------- | ---------- | ---------- | ---------- | ---------- | ----------- | ---------- | ---------- |
> | GCN             | 0.9179     | 0.9290     | 0.9514     | 0.9240     | 0.5175      | 0.5260     | 0.5585     |
> | GAT             | 0.9383     | 0.9337     | 0.9507     | 0.9334     | 0.6077      | 0.3905     | 0.5623     |
> | MLP+hybrid attn | **0.9556** | **0.9415** | **0.9581** | **0.9603** | **0.8117**  | **0.5983** | **0.5809** |
>
> The experimental results indicate that simple convolution(GCN) is ineffective due to the over-smoothing problem, where convolution tends to make anomalies indistinguishable. Although traditional attention mechanisms(GAT) can yield competitive results in some cases, the overall performance doesn't match GADAM. This discrepancy arises because, at the beginning of training in the second stage, the encoder as well as $H_{global}$ is not fully optimized. Node-node attention relies solely on feature similarity, offering less effective guidance in the early stages despite its finer granularity. In contrast, our hybrid attention, which integrates pre-attn based on anomaly scores and post-attn based on feature similarity through a dynamic weighted sum, has proven to be a highly effective design, as demonstrated in Sec 4.5.2.
>
> If we have misunderstood your concerns, feel free to let us know!

---

> ### Author Response · Authors · 2023-11-17
> **Response to reviewer deYB, Part III**
>
> **W6.** Issues about baseline performance reported in this paper, particularly in comparison to the AnomalyDAE paper. The AUC are 97.81 (BlogCatalog), 90.05 (ACM) in the original paper; whereas in this paper, they are 76.58 (BlogCatalog), 75.16 (ACM).
>
> Thank you for bringing up this concern! We've thoroughly reviewed the original AnomalyDAE paper and its associated source code[3]. However, it appears that the anomaly injection method is not clearly outlined in the article, and the provided source code only includes the BlogCatalog dataset with anomaly labels. To ensure a fair and comprehensive comparison, we utilized PyGOD[4], an open library from the NIPS'22 benchmark for unsupervised graph anomaly detection. We applied this library uniformly across all datasets to generate anomalies, striving for consistency and fairness.
>
> Simultaneously, we conducted experiments using the parameter configurations specified in the original AnomalyDAE paper. The obtained results align with those reported in the updated work[5]. Regrettably, due to the lack of clarity on how AnomalyDAE injects anomalies and the absence of support for other datasets in its source code, we couldn't replicate experiments on additional datasets using the provided code. We made diligent efforts to ensure the fairness of our experiments through PyGOD, and observed performance variations in AnomalyDAE may be dataset-dependent.
>
> **W7.** How to decide topk% included in the second stage. How does the selection of this percentage affect the results?
>
> Thanks for brining this up! For both $k_{nor}$ and $k_{ano}$, the common setting principle is to include as many correct supervisory signals as possible.
>
> + $k_{nor}$ determines the size of the high-confidence normal set $V_{n}$。Given that normal nodes constitute the majority of the dataset,  $V_{n}$ generally includes a very small proportion of true anomalies. Consequently,  $k_{nor}$ can be set as long as it ensures an adequate supply of supervisory signals,  and we set it to a larger value in our experiments. We also found in our implementation that the setting of $k_{nor}$ has minimal impact on the model's performance.
> + $k_{ano}$ determines the size of the high-confidence abnormal set $V_a$, it seeks to include as many true anomalies as possible and exclude normal nodes. So $k_{ano}$ is ideally set to be as close as possible to, but not substantially larger than, the proportion ($p$) of anomalies in the datasets. Intuitively, when $k_{ano}>p$, a continuous increase in $k_{ano}$ will cause more normal nodes to be marked as anomalous, thus provide more false pseudo-labels for the second stage, and lead to model's performance drop. Importantly, we kindly emphasize that **we do not rely on the true anomaly proportion of the dataset as a priori when setting $k_{ano}$.** We just empirically set $k_{ano}$ within a reasonable interval, typically no greater than 5. Ablation studies conducted in Sec 4.5.1 also demonstrate the robustness of GADAM to different $k_{ano}$ values.
>
> **W8.** The paper is missing some unsupervised graph anomaly detection papers。
>
> We appreciate your thorough review and a wealth of related works you've provided. We carefully examined these works you mentioned and will introduce each method systematically:
>
> + **AEGIS[6]:** AEGIS mainly focuses on inductive learning, including an anomaly-aware GNN layer and gives the model the ability to detect new anomalies by generative adversarial learning. Unfortunately, AEGIS is not open sourced and unavailable through PyGOD, a fair comparison may not be feasible.
> + **VGOD[7]:**  VGOD designs two modules for structural and contextual anomalies respectively, which belongs to the same domain as our work. In the following section, we will conduct a detailed comparison with VGOD.
> + **AHEAD[8]:** AHEAD is designed for anomaly detection in heterogeneous graphs. Therefore its research has little relevance to us, and its open source code is specifically implemented for IMDB dataset, so we cannot include AHEAD in our baseline.
> + **GraphBEAN[9]:** GraphBEAN is designed for bipartite node-and-edge-attributed graphs, and detects anomalous nodes and edges simultaneously. Taking the example of consumer-purchase-product graph, GraphBEAN is designed to detect edges representing anomalous transactions, and users with anomalous behavior. Thus its research area has little overlap with our work, and we won't consider it as a baseline in our study.
> + **ACT[10]:** ACT focuses on cross-domain graph anomaly detection (CD-GAD), which describes the problem of detecting anomalous nodes in an unlabeled target graph using auxiliary, related source graphs with label nodes. Recognizing the specific requirements for auxiliary source graphs,  we agree that it's not suitable for inclusion in our baseline.

---

> ### Author Response · Authors · 2023-11-17
> **Response to reviewer deYB, Part IV**
>
> **GADAM vs. VGOD**
>
> The anomaly detection module of VGOD is designed specifically for two types of anomalies. It uses autoencoder to reconstruct the nodes and the reconstruction error is used to detect the contextual anomaly. At the same time, it uses the variance of the node's neighborhood features as a measure of the structural anomaly score, and finally the two anomaly scores are jointly used as the criteria for anomaly detection. Therefore, in order to explore whether VGOD is also applicable to real datasets containing organic anomalies, we extended our evaluation to include the Disney dataset, and the statistic of Disney is as follows:
>
> |        | #Nodes | #Edges | Degree | #Anomalies | Ratio |
> | ------ | ------ | ------ | ------ | ---------- | ----- |
> | Disney | 124    | 335    | 2.7    | 6          | 4.8%  |
>
> We performed extensive comparative experiments across seven datasets, where * denotes datasets with organic anomalies. Except for Disney, the other datasets are the ones utilized in our paper, and their statistics are provided in Appendix A.2. The results, evaluated under various metrics, are presented below:
>
> **AUC-ROC**
>
> |       | Cora       | Citeseer   | Pubmed     | BlogCatalog | Books*     | Reddit*    | Disney*    |
> | ----- | ---------- | ---------- | ---------- | ----------- | ---------- | ---------- | ---------- |
> | VGOD  | 0.9503     | **0.9845** | **0.9813** | 0.796       | 0.373      | 0.523      | 0.387      |
> | GADAM | **0.9556** | 0.9415     | 0.9581     | **0.8122**  | **0.5983** | **0.5809** | **0.8178** |
>
> **recall@k**
>
> |       | Cora       | Citeseer | Pubmed    | BlogCatalog | Books*     | Reddit*    | Disney*    |
> | ----- | ---------- | -------- | --------- | ----------- | ---------- | ---------- | ---------- |
> | VGOD  | 0.593      | **0.76** | **0.555** | 0.313       | 0          | 0.025      | 0          |
> | GADAM | **0.7299** | 0.712    | 0.462     | **0.3667**  | **0.0143** | **0.0699** | **0.1667** |
>
> **average precision**
>
> |       | Cora      | Citeseer | Pubmed   | BlogCatalog | Books*     | Reddit*    | Disney*    |
> | ----- | --------- | -------- | -------- | ----------- | ---------- | ---------- | ---------- |
> | VGOD  | 0.695     | **0.84** | **0.56** | 0.212       | 0.016      | 0.035      | 0          |
> | GADAM | **0.728** | 0.7512   | 0.4264   | **0.296**   | **0.0279** | **0.0481** | **0.1695** |
>
> Experimental results show that VGOD is a very strong baseline on some generative datasets (Citeseer, Pubmed), but its performance on real datasets containing organic anomalies has a dramatic drop. Especially on Books and Disney, its performance is not good than random guess. We attribute this observation to VGOD being specifically tailored for synthetic anomalies, making its poor performance on real-world datasets. In contrast, our model excels by capturing two types of anomaly scores from both local and global perspectives, showcasing superior generalization capabilities and practical utility for real-world applications.
>
> We sincerely thank you for the valuable related works, we will integrate the results into the later revision, which will help us to achieve a great improvement of our paper!
>
> **Reference**
>
> [1] Anomaly detection on attributed networks via contrastive self-supervised learning (TNNLS 2021)
>
> [2] Anemone: graph anomaly detection with multi-scale contrastive learning (CIKM 2021)
>
> [3] https://github.com/haoyfan/AnomalyDAE
>
> [4] BOND: Benchmarking Unsupervised Outlier Node Detection on Static Attributed Graphs (NIPS 2022)
>
> [5] ComGA: Community-Aware Attributed Graph Anomaly Detection (WSDM 2022)
>
> [6] Inductive anomaly detection on attributed networks. (IJCAI 2021)
>
> [7] Are we really making much progress in unsupervised graph outlier detection? Revisiting the problem with new insight and superior method. (arxiv 2022)
>
> [8] Ahead: A triple attention based heterogeneous graph anomaly detection approach (2023)
>
> [9] Interaction-Focused Anomaly Detection on Bipartite Node-and-Edge-Attributed Graphs (IJCNN 2023)
>
> [10] Cross-domain graph anomaly detection via anomaly-aware contrastive alignment (AAAI 2023)

---

> > ### Comment · Reviewer_deYB · 2023-11-23
> >
> > Thanks the authors for the detailed response.
> > I do not have additional concerns.
> >
> > For the additional papers I listed, even though they may not totally align with this paper direction (and hence not suitable for baseline), I still suggest citing the papers, to provide the reader a broader overview of the research on unsupervised graph anomaly detection.

---

> > > ### Author Response · Authors · 2023-11-23
> > > **Response to reviewer deYB**
> > >
> > > Thank you for your valuable time and constructive suggestions! We will incorporate these papers into our related work as space permits.

---

> ### Author Response · Authors · 2023-11-23
> **Follow-up response**
>
> Based on your valuable suggestions, we have integrated the related works you suggested into Sec. 5 of the revision (detailed in Appendix B.4). We are pleased that our detailed response has helped address your concerns. We sincerely hope that these enhancements will contribute to achieving a higher score!

---

### Official Review · Reviewer_WYWU · 2023-11-09

**Soundness:** 3 good
**Presentation:** 3 good
**Contribution:** 2 fair
**Rating:** 5
**Confidence:** 4

**Summary:**

This paper addresses the problem of unsupervised node-level anomaly detection in a graph through an adaptive message passing framework. The paper rightly motivates the fact that message passing in GNNs results in local anomaly signal loss. The authors have proposed an approach to obtain local and global anomaly scores through MLP and hybrid attention based adaptive message passing. The paper has conducted experiments on anomaly detection on real world datasets with both injected and ground truth anomalies, and used a good number of baseline approaches from the literature.

**Strengths:**

1. The idea of using local anomaly score to compute the global anomaly score is interesting.
2. The paper uses a good number of baseline algorithms from the literature.

**Weaknesses:**

1. The problem formulation seems to be problematic. Authors have mentioned "Structural anomalies are densely connected nodes in contrast to sparsely connected regular nodes". What about the networks that are dense by nature? Moreover, if density is the only criteria for being a structural anomaly, why can't we use degree of a node as the metric to find the structural anomalies? Are we over-complicating the solution of a relatively simple problem?

2. I am not convinced with the proposed solution. s^local (in Eq. 6) can only capture contextual anomalies, but not the structural anomalies because of the L2 normalization in Eq. 3. Where exactly are we capturing the structural anomalies in this framework?

3. What is the difference between H_local and H_global? They seem to be computed in the same way through Eq. 3. Why are we calling the second one as "global"?

4. Some of the claims in the paper should have been explained more. Can you please elaborate the reason of "as the training advances, the influence of pre-attention gradually diminishes, while the opposite for post-attention".

5. In Table 1, overall anomaly detection performance is reported. I am curious to see the performance on contextual and structural anomalies separately.

6. As mentioned before, degree centrality seems to be a good metric to capture the structural anomalies. Can you please include this heuristic as a baseline to compare with?

7. How do you capture the nodes which are both contextual and structural anomalies? Also, will your approach be able to capture "Combined Outliers" (or combined anomalies) as introduced in "Outlier Aware Network Embedding for Attributed Networks" (AAAI-2019)?

**Questions:**

Please see the questions above.

***********************************

After Rebuttal: The authors have tried to address all the concerns I had. I am changing my score from 3 to 5. I believe the paper needs a major revision to incorporate the recommended changes and the new experiments. I don't mind if this paper gets accepted in ICLR 2024 with those changes.

---

> ### Author Response · Authors · 2023-11-17
> **Response to reviewer WYWU, Part I**
>
> Thanks for your insightful review. We provide detailed answers to your questions as follows, and we hope that our response can address your concerns.
>
> **W1.** Thanks for brining up this fundamental question! We will address your concerns one by one:
> > (1) Definition of structural anomaly
>
> The definition of structural anomaly in our paper aligns with the latest benchmark from NIPS'22[1]. Notably, this definition is widely used in the field of unsupervised graph anomaly detection and is evident in the majority of existing works[2,3,4,5]. Structural anomaly is prevalent in the real world and is characterized by **nodes which are densely connected with abnormal links**. Examples range from members of organized fraud gangs, people from different communities but closely connected to each other, and even papers that engage in malicious citations. In our work, we adopt the anomaly injection method used in previous papers[1,2,3,4,5] to generate densely and abnormally connected nodes as structural anomalies (see Appendix A.1). While we acknowledge that solely emphasizing on "densely connected" can lead to ambiguities in the understanding of structural anomalies, and we will add a new term for "abnormal links" in a later revision.
>
> > (2) What if the networks are dense by nature?
>
> As discussed earlier, **structural anomalies are essentially clusters of densely and abnormally linked nodes**. Therefore, there is no inherent conflict between the graph itself being dense and the definition of structural anomalies. In a dense graph, normal nodes are normally connected to each other, while structural anomalies have abnormal connections to each other. As we introduced in Appendix A.1, when generating structural anomalies, we randomly select $m$ (e.g. $m=15$) nodes and connect them to each other. The randomly selected nodes come from different communities, thus creating densely and abnormally linked structural anomalies, which is consistent with the definition of structural anomalies.
>
> > (3) Can we use node degrees to find structural anomalies？
>
> This is indeed an interesting idea. However, as previously discussed, employing node degree is not aligned with the essence of detecting structural anomalies, since the degree does not reflect the abnormal links of nodes. For instance, in networks like Twitter's following network or paper citation networks such as Cora, it is clear that we cannot simply mark accounts with high number of followers or highly cited papers as anomalies, even though they have significantly higher node degrees compared to other nodes. We will delve deeper into validating the effectiveness of utilizing node degree for structural anomaly detection in response to W6.
>
> > (4) Are we over-complicating the solution of a relatively simple problem?
>
> Following the discussion above, we don't consider the detection of structural anomalies to be overly complicated. Utilizing node degree only exploits the phenomenon of being densely connected and can easily fail in graphs with high average node degrees, as we will demonstrate in response to W6. In contrast, our model takes a principled approach to anomaly detection by capturing two types of anomaly signals from both local and global perspectives. The anomaly signals are propagated through dense links in the cluster of abnormal nodes by adaptive message passing in the second stage. This process enhances the anomaly scores of structural anomalies as a whole, and we believe this approach possesses greater generalization capabilities and is more in line with the essence of anomaly detection.
>
>
>  **W2.** Thank you for raising this point.
>
>  Firstly, the anomaly signal in the first stage is derived from the local inconsistency between nodes and their neighbors. L2 normalization is applied to normalize node features, facilitating the use of cosine similarity as a consistency discriminator. However, **this doesn't lead to the failure of detecting structural anomalies**. It's worth noting that the neighbors of some structural anomalies may include normal nodes(e.g. node 7 in Figure 2 of our paper). Consequently, these structural anomalies can exhibit high local inconsistency, making them detected in the first stage. Hence, $S^{local}$ remains effective in detecting structural anomalies with this characteristic.
>
> In addition to the structural anomalies described above that are captured, in the second stage, we enhance structural anomaly detection through adaptive message passing. Specifically, structural anomalies identified in the first stage act as supervisory signals in the second stage, guiding the model to push them away from the global normal context. The hybrid attention mechanism plays a crucial role in this process by amplifying the message passing of these structural anomalies in a cluster. Consequently, other structural anomalies align with these captured nodes and move away from the global normal context, thereby contributing to an improved overall cluster anomaly score.

---

> > ### Comment · Reviewer_WYWU · 2023-11-20
> > **Response to Authors**
> >
> > Dear Authors,
> >
> > Thank you for your reply.
> >
> > "In a dense graph, normal nodes are normally connected to each other, while structural anomalies have abnormal connections to each other" - Can you please define "abnormality" in the context of structural anomalies? In Fig. 1 of the paper, I can see structural anomalies are densely connected among themselves. But how to even manually understand which links are normal and which are abnormal without the help of node attributes? In this discussion, I want to keep node attributes separate since they are captured through contextual anomalies. Also, since these structural anomalies are densely connected among themselves, their edges to other nodes in the graph can be somewhat insignificant. Is there a way we can separate them out from "genuine" densely connected nodes in the graph without using node attributes?

---

> > > ### Author Response · Authors · 2023-11-20
> > > **Response to reviewer WYWU**
> > >
> > > We appreciate your attention to "structural anomalies." We will answer your questions further:
> > >
> > > > (1)Can you please define "abnormality" in the context of structural anomalies?
> > >
> > > The "abnormality" of structural anomalies in our model can be understood through two key properties:
> > >
> > > + **Connections between nodes belonging to different communities (Property 1):** Nodes forming connections between distinct communities contribute to the structural abnormality.
> > >
> > > + **Dense connections (Property 2):** Densely connected nodes in contrast to sparsely connected regular nodes.
> > >
> > > It's essential to clarify that, the definition of structural anomaly is a holistic concept[1]. These two properties together determine whether a cluster is anomalous or not, and further, whether a node in that cluster is structural anomaly or not. Note that our structural anomaly generation method aligns with these two properties, as detailed in our response to W1.
> > >
> > > > (2)How to even manually understand which links are normal and which are abnormal without the help of node attributes?
> > >
> > > The diagram in Fig.1 only shows the structural anomaly densely connected for clarity and ease of understanding, and in fact these nodes will be distributed in different communities. Without using the node attribute, we can only determine whether a node satisfies Property 2, not whether it satisfies Property 1, and therefore cannot achieve a comprehensive detection for structural anomalies.
> > >
> > > > (3)Is there a way we can separate them out from "genuine" densely connected nodes in the graph without using node attributes?
> > >
> > > In graphs that are inherently dense, if the node attribute is not used, then both properties in (1) will fail. Thus to the best of our knowledge, no such method exists in the unsupervised graph anomaly detection. It's worth noting that most of existing methods[1,2,3,4] rely on node attributes for structural anomaly detection. Many of these methods reconstruct the edges of the graph based on node attributes and utilize the reconstruction error to identify structural anomalies. Since structural anomalies come from different communities with diverse attributes, it is difficult to achieve a better reconstruction and thus be identified as anomalies. In our approach, we capitalize on Property 1 in the first stage, mining local inconsistency to identify certain structural anomalies. Furthermore, we leverage the densely connected property in the second stage for further enhanced detection, as detailed in our response to W2.
> > >
> > > We hope that our explanations have provided you with a better understanding of the issues, and we welcome any other questions and further discussions with us.
> > >
> > > **Reference**
> > >
> > > [1] BOND: Benchmarking Unsupervised Outlier Node Detection on Static Attributed Graphs (NIPS 2022)
> > >
> > > [2] Deep anomaly detection on attributed networks(SDM 2019)
> > >
> > > [3] ComGA: Community-Aware Attributed Graph Anomaly Detection (WSDM 2022)
> > >
> > > [4] Reconstruction Enhanced Multi-View Contrastive Learning for Anomaly Detection on Attributed Networks (IJCAI 2022)

---

> ### Author Response · Authors · 2023-11-17
> **Response to reviewer WYWU, Part II**
>
> **W3**. What is the difference between H_local and H_global? Why are we calling the second one as "global"?
>
> Indeed, both $H_{local}$ and $H_{global}$ are calculated in the same way through Eq.(3), but as we claimed in  Sec. 3.2.1, $H_{global}$ employs a new MLP as an encoder.
>
> To elaborate, in the first stage, we utilize one MLP on the raw node features to obtain $H_{local}$, employing the local inconsistency of the nodes as anomaly signals and learning $H_{local}$ through Eq.(1). While in the second stage, we deploy another MLP on the raw features to get $H_{global}$, mining the anomaly signals from the consistency of the nodes and global normal context, and learning $H_{global}$ through Eq.(13). We refer to the second one as "global" from the perspective of anomaly signaling.
>
> **W4**.  Can you please elaborate the reason of "as the training advances, the influence of pre-attention gradually diminishes, while the opposite for post-attention"?
>
> Thanks for brining this up. We will explain the reasons for this in terms of formula definitions and design intuitions, respectively:
>
> + As defined in Ep.(12), pre-attention and post-attention regulate the weights by a weight coefficient $\beta_t$ that varies with the training step. In this process the weight of pre-attn is gradually decreased while the weight of post-attn is increased through $\beta_t$.
> + This is a special design with the following considerations: (1) At the beginning of training in the second stage,  $H_{global}$ is not fully optimized, and pre-attn serves a reliable criterion to judge whether a node should align with its neighbors. However, we consider that the local anomaly scores in the first stage may contain inherent errors. For example, it fails to detect certain structural anomalies with high local consistency. Gradually reducing the weight of pre-attn aims to prevent inherent errors in local anomaly scores from continuously misdirecting attention for nodes. (2) As training progresses, nodes adaptively align with their neighbors through message passing involving hybrid attention. Therefore, we increase the weight of post-attn based on feature similarity as a complementary and corrective measure to pre-attn.
> + In the ablation experiments in Section 4.5.3, we also examined various attentional mechanisms. The results demonstrate that the dynamic sum approach exhibits a substantial advantage over other alternatives.
>
> The explanations provided above are also articulated in our paper, accompanied by visualizations of the attention coefficients in Appendix E. These visualizations demonstrate that hybrid attention effectively reduces the message passing of contextual anomalies while enhancing them for normal nodes and structural anomalies with their surroundings. We hope that our explanations could help you further understand our design motivations!
>
> **W5.** I am curious to see the performance on contextual and structural anomalies separately.
>
> Thanks for your valuable comment! To assess the efficacy of our model in detecting two types of anomalies, we selected three benchmark datasets for experimentation. **Cora** serves as a small, regular dataset with a low average node degree. **ACM**, on the other hand, represents a larger dataset with a substantial number of anomalies. Finally, **BlogCatalog** is characterized by a large average node degree, presenting a greater challenge for the detection of structural anomalies. Additionally, we chose representative models from the baseline families—DOMINANT, CoLA, Sub-CR and ComGA—for comparative experiments. The experimental results under the AUC-ROC metric are as follows:
>
> |          | Cora       |            | ACM        |            | BlogCatalog |            |
> | -------- | ---------- | ---------- | ---------- | ---------- | ----------- | ---------- |
> |          | contextual | structural | contextual | structural | contextual  | structural |
> | DOMINANT | 0.7173     | 0.9312     | 0.7045     | 0.8182     | 0.7371      | 0.7664     |
> | CoLA     | 0.8934     | 0.8213     | 0.9103     | 0.8325     | 0.8585      | 0.7325     |
> | ComGA    | 0.8173     | 0.9052     | 0.7525     | 0.8989     | 0.8294      | 0.7679     |
> | Sub-CR   | 0.9057     | 0.9293     | 0.7067     | 0.7913     | 0.8267      | 0.7573     |
> | GADAM    | **0.9416** | **0.9929** | **0.9586** | **0.9721** | **0.9094**  | **0.7763** |
>
> The results show that:
>
> + Most of the baselines struggles to balance the detection of both types of anomalies, while GADAM achieves the best.
> + On the two sparser datasets, Cora and ACM, GADAM significantly outperforms baselines in structural anomaly detection. Even on the denser BlogCatalog dataset, GADAM continues to outperform all baselines, providing further validation of the effectiveness for structural anomaly detection. This outcome aligns with our model designs described in W2.
>
> We intend to incorporate these valuable results into our later revisions, thanks again for your valuable comment!

---

> ### Author Response · Authors · 2023-11-17
> **Response to reviewer WYWU, Part III**
>
> **W6.** Can you please include degree as a baseline to compare with?
>
> Certainly. Given that degree is only capable of detecting structural anomalies, we evaluated its performance **specifically on structural anomalies** using the benchmark datasets mentioned in our paper. We have also provided the average node degree (avg_d) for each dataset, and listed the datasets in increasing avg_d to offer more clarity. The results, measured under the AUC-ROC metric, are as follows:
>
> |        | Citeseer(avg_d = 1.4) | Pubmed(avg_d = 2.3) | ACM(avg_d = 4.4) | ogbn-Products(avg_d = 25.3) | BlogCatalog(avg_d = 33.1) |
> | ------ | ------------------------- | ----------------------- | -------------------- | ------------------------------- | ----------------------------- |
> | Degree | 0.9919                    | 0.9421                  | 0.9264               | 0.6397                          | 0.64                          |
> | GADAM  | **0.9956**                | **0.9718**              | **0.9721**           | **0.7963**                      | **0.7763**                    |
>
> As evident from the experimental results:
>
> + Degree is indeed a good metric for detecting structural anomalies in certain datasets but exhibits poor performance in datasets characterized by high average degrees (e.g., ogbn-Products, BlogCatalog). This limitation underscores the metric's challenges in achieving robust generalization across diverse datasets.
> + GADAM consistently outperforms Degree across all datasets, with the disparity being particularly pronounced in datasets featuring higher average degrees. This robust performance stems from GADAM's design, rooted in the fundamental principles of anomaly detection, enabling stronger generalization across diverse datasets.
>
> We acknowledge that the effectiveness of using degree as a metric for detecting structural anomalies may be contingent on the limitations of current anomaly generation methods. Designing more realistic and flexible synthetic outlier generation in graphs is also a critical challenge[1]. Our work, while based on a general anomaly injection approach, strives to align with the essence of anomaly detection. We sincerely hope that our underlying motivations resonate with your understanding!
>
> **W7.** How do you capture the nodes which are both contextual and structural anomalies?
>
> Thanks for brining this up! Intuitively, our framework is poised to effectively capture nodes that exhibit both contextual and structural anomalies (mixed anomalies for short). The first stage of GADAM captures well the contextual abnormal features exhibited by mixed anomalies. In the second stage, GADAM facilitates the absorption of anomalous signals from the neighboring structural anomalies through adaptive message passing, further enhancing the detection of mixed anomalies.
>
> Moreover, after a thorough examination of the mentioned paper (AAAI-2019), we confirm that "combined anomalies" align with our concept of mixed anomalies. To further address your concern, we generated a certain number of mixed anomalies on the five benchmark datasets. Specifically, we randomly selected $q$ structural anomalies from the original datasets in our paper, and employed the contextual anomaly injection method outlined in Appendix A.1 to substitute features for these nodes, thereby creating $q$ mixed anomalies. Our evaluation utilized the representative baselines mentioned in W5, and the performance of mixed anomaly detection under the AUC-ROC metric is as follows:
>
> |          | Cora (q=15) | Citeseer(q=15) | Pubmed(q=60) | ACM(q=60) | BlogCatalog(q=30) |
> | -------- | -------------- | ------------------ | ---------------- | ------------- | --------------------- |
> | DOMINANT | 0.8354         | 0.8498             | 0.7904           | 0.7448        | 0.7723              |
> | CoLA     | 0.8976         | 0.8928             | 0.9618           | 0.7909        | 0.7965              |
> | ComGA    | 0.8828         | 0.9335             | 0.9255           | 0.8433        | 0.8275                |
> | Sub-CR   | 0.9236         | 0.9400             | 0.9810           | 0.7518        | 0.7963                |
> | GADAM    | **0.9970**      | **0.9994**         | **0.9863**       | **0.9884**    | **0.9685**            |
>
> The results indicate that GADAM exhibits a strong ability to detect mixed anomalies across all datasets, aligning with the motivation behind the model design. We will incorporate these findings into a subsequent revision. Thanks for your insightful review!
>
> **Reference**
>
> [1] BOND: Benchmarking Unsupervised Outlier Node Detection on Static Attributed Graphs(NIPS 2022)
>
> [2] Deep anomaly detection on attributed networks(SDM 2019)
>
> [3] Anomaly detection on attributed networks via contrastive self-supervised learning(TNNLS 2021)
>
> [4] Reconstruction Enhanced Multi-View Contrastive Learning for Anomaly Detection on Attributed Networks(IJCAI 2022)
>
> [5] Generative and contrastive self-supervised learning for graph anomaly detection(TKDE 2021)

---

> > ### Comment · Reviewer_WYWU · 2023-11-20
> > **Reviewer Response**
> >
> > I thank the authors for posting the results on contextual and structural anomalies separately and other experiments they have conducted to answer the doubts I have. I also request them to incorporate mixed (or combined) anomalies in the paper if space permits.

---

> > > ### Author Response · Authors · 2023-11-20
> > > **Response to reviewer WYWU**
> > >
> > > We are glad that the results of our experiments have addressed your concerns, and we have integrated the results into Appendix G in a revision. Thanks again for your valuable review, which helped us to improve our paper considerably!

---

### Author Response · Authors · 2023-11-21
**Gentle Reminder.**

Dear Reviewers,

We extend our sincere gratitude for your invaluable reviews. In light of your constructive feedback, we have:

(1) Conducted additional experiments. These include conducting additional experiments, such as a detailed exploration of the model's performance on distinct anomaly types (refer to Appendix G in the revision) and incorporating three representative baselines beyond the realm of deep learning and neural networks (see our response to reviewer 6vV6, Part I). Moreover, we have conducted additional experiments in each response, specifically tailored to address the raised concerns.

(2) Provided the **theoretical guarantee** of the second stage of GADAM, see Appendix F in the revision;

(3) Carefully prepared our response to your questions.

We hope that the additional results could address your concerns about the performance of GADAM. We also hope that the further revision could enhance the clarity of our problem settings, motivations, and methods. As the author response period is coming to an end, we would appreciate it if you could consider our response and we are more than willing to address any further comments or questions.
Once again, we thank the reviewers for the valuable feedback, which has undoubtedly contributed to improving the quality of our work.

---

### Meta-Review · Area_Chair_nFmt · 2023-12-06

**Metareview:**

This paper aims to develop new methods to detect anomaly in graphs. It is observed that the message passing scheme of GNNs is often odd with the anomaly detection method called local inconsistency mining (LIM). The paper then proposes a two-stage framework that performs LIM and message passing separately. Both structural anomaly and contextual anomaly are considered to boost the performance of the algorithm. There is a comprehensive set of evaluations of the models on benchmark data sets, showing that the new approach is often effective.

**Strengths**
- The paper is clearly written, and is very well motivated. Notations are easy to track, and are always clearly explained.
- The proposed two-stage framework is natural and easy to follow. The main steps are elaborated in a section.
- The experiments are comprehensive. Authors further clarified settings in the rebuttal.

**Weaknesses**
- Some reviewers raised a concern that the evaluation results of some baseline methods are reported lower than they were in the original papers. While authors addressed the concern in the rebuttal (and appears a reasonable justification), authors are strongly advised to clarify in their revision. In particular, authors applied different anomaly generation paradigm but sticked with the original hyper-parameters of these baseline methods, which may lead to unfair comparison as such default parameters can be sub-optimal under the current pipeline.

**Suggestions to authors**
- Authors are suggested to clarify the experimental settings.
- Authors are strongly suggested to well-tune the hyper-parameters of the baseline methods.
- Authors are suggested to incorporate other reviewer feedback into the revision.

**Justification For Why Not Higher Score:**

The paper may need a nontrivial revision. Some baseline methods need to be re-evaluated.

**Justification For Why Not Lower Score:**

The approach is well-motivated and easy to follow. Strengths outweigh weaknesses.

---

### Decision · Program_Chairs · 2024-01-16

Accept (poster)